# Comprehensive Time-Course Transcriptome Reveals the Crucial Biological Pathways Involved in the Seasonal Branch Growth in Siberian Elm (*Ulmus pumila*)

**DOI:** 10.3390/ijms241914976

**Published:** 2023-10-07

**Authors:** Luo-Yan Zhang, Cheng Yang, Zhi-Cheng Wu, Xue-Jie Zhang, Shou-Jin Fan

**Affiliations:** Key Lab of Plant Stress Research, College of Life Science, Shandong Normal University, No. 88 Wenhuadong Road, Ji’nan 250014, China; zhangluoyan@sdnu.edu.cn (L.-Y.Z.); 2020020799@stu.sdnu.edu.cn (C.Y.); 2021020857@stu.sdnu.edu.cn (Z.-C.W.)

**Keywords:** *Ulmus pumila*, time-course transcriptome, seasonal branch growth, xylan synthesis, cold stress responding

## Abstract

Timber, the most prevalent organic material on this planet, is the result of a secondary xylem emerging from vascular cambium. Yet, the intricate processes governing its seasonal generation are largely a mystery. To better understand the cyclic growth of vascular tissues in elm, we undertook an extensive study examining the anatomy, physiology, and genetic expressions in Ulmus pumila. We chose three robust 15-year-old elm trees for our study. The cultivars used in this study were collected from the Inner Mongolia Autonomous Region in China and nurtured in the tree farm of Shandong Normal University. Monthly samples of 2-year-old elm branches were taken from the tree from February to September. Marked seasonal shifts in elm branch vascular tissues were observed by phenotypic observation: In February, the cambium of the branch emerged from dormancy, spurring growth. By May, elms began generating secondary xylem, or latewood, recognized by its tiny pores and dense cell structure. From June to August, there was a marked increase in the thickness of the secondary xylem. Transcriptome sequencing provides a potential molecular mechanism for the thickening of elm branches and their response to stress. In February, the tree enhanced its genetic responses to cold and drought stress. The amplified expression of *CDKB*, *CYCB*, *WOX4,* and *ARF5* in the months of February and March reinforced their essential role in the development of the vascular cambium in elm. Starting in May, the elm deployed carbohydrates as a carbon resource to synthesize the abundant cellulose and lignin necessary for the formation of the secondary wall. Major genes participating in cellulose (*SUC* and *CESA* homologs), xylan (*UGD*, *UXS*, *IRX9*, *IRX10,* and *IRX14*), and lignin (*PAL*, *C4H*, *4CL*, *HCT*, *C3H*, *COMT,* and *CAD*) biosynthetic pathways for secondary wall formation were up-regulated by May or/and June. In conclusion, our findings provided a foundation for an in-depth exploration of the molecular processes dictating the seasonal growth of elm timber.

## 1. Introduction

Over 90% of the Earth’s terrestrial ecosystem’s biomass is constituted by woody material [1,2]. The continuous vertical and radial expansion of trees is a hallmark of their growth throughout their lifespan. The growth in the diameter of trees is attributed to the vascular cambium’s operations, which yield secondary xylem and phloem [3,4]. Understanding the cyclical nature of wood growth is pivotal for insights into timber development and biomass aggregation. Presently, studies focusing on this domain primarily involve poplar trees and gymnosperms [5,6]. In the cooler northern climates, the vascular progression of timber species follows a pattern of active growth interspersed with periods of inactivity [6,7]. In temperate trees, the dormant cambium springs to life at the outset of spring, leading to cellular splitting and swift enlargement, where the creation of the primary cell wall takes precedence. As the growth season advances, an increasing number of xylem cells partake in the formation of the secondary wall and undergo lignification, culminating in a scheduled cellular demise come fall [3,4]. Yearly growth rings typically exhibit two contrasting sections: the earlywood (EW) and the latewood (LW) [4]. EW emerges during spring and the initial phase of summer when climatic conditions, such as temperature, precipitation, and daylight duration, are conducive to swift growth. In contrast, LW is generated predominantly from the heart of summer through to fall [8].

The vascular cambium is responsible for secondary growth, creating wood (or secondary xylem) internally and bast (or secondary phloem) externally through a precise, two-directional process that involves synchronized cell division and differentiation [3,4]. In the plant Arabidopsis, the initiation of the vascular cambium and the subsequent growth and differentiation of its derivative cells necessitate the harmonization of various signals. These signals encompass auxin, cytokinin (CK), brassinosteroid (BR), gibberellin (GA), ethylene (ET), and TDIF-PXY [4]. Within pre-procambial strands, in reaction to auxin, ARF5 is activated. This boosts the count of initial vascular cells by promoting the expression of the auxin efflux carrier gene *PIN1* [9]. *WOX4* is crucial for the auxin-mediated enhancement of cambium function and is believed to be a primary driver of vascular cambium division [10,11]. In the case of *Populus*, the rising temperatures at the onset of spring potentially trigger the transcription of *CDKB* and *CYCB* analogs within the cambium zone, a vital step for the division of cambium cells [12].

The primary constituents of secondary xylem are deceased cells with fortified cell walls abundant in cellulose, hemicelluloses, and lignin [13,14,15]. This structure not only delivers mechanical reinforcement to the plant but also facilitates the movement of water and minerals. Secondary cell walls (SCWs) evolve from xylem cells, constituting a significant proportion of plant lignocellulosic biomass [13,16]. This biomass is viewed as a sustainable source for bioenergy production. Comprehensive biochemical, genetic, and molecular analyses have unveiled that myriad genes partake in crafting SCW components [13,15]. Cellulose is formed in the plasma membrane by cellulose synthase (CESA) complexes [17]. Three notable SCW CESA genes in Arabidopsis, namely *CESA4*/*IRX5*, *CESA7*/*IRX3*, and *CESA8*/*IRX1*, are paramount for plant maturation. Altering these genes culminates in significantly diminished cellulose content and SCW density, causing weaker stem rigidity and a distorted xylem appearance [13,18]. Xylan synthesis occurs in the Golgi apparatus, which is inclusive of its β-1,4-xylan backbone, sugar attachments, and acetyl clusters. These components are then dispatched to the cell walls via vesicles. Scientific investigations, both genetic and biochemical, have ascertained that a xylan synthase cluster orchestrates the crafting of the β-1,4-xylan backbone. This complex amalgamates family GT47 proteins (IRX10/IRX10L) and two distinct sets of family GT43 proteins (IRX9/IRX9L and IRX14/IRX14L) [19,20].

The creation of cell wall components is meticulously governed by complex networks of transcription factors (TFs) [13]. Predominantly, the TFs involved in the SCW regulatory matrix stem from two primary families: the NAC TFs and the MYB TFs [21,22]. Depending on their role in the regulatory hierarchy, these TFs are categorized into three distinct tiers. Tier 1 TFs, which include MYB TFs, such as *MYB20*, *MYB69*, *MYB79*, *MYB85*, *MYB58*, and *MYB63*, latch onto the cis-elements directly influencing the transcription of foundational genes. Tier 2 TFs, encompassing *MYB46*, *MYB83*, *MYB55*, *SND3*, and *XND1*, have a dual role; they not only control the expression of Tier 1 TFs but also govern foundational genes. Finally, Tier 3 TFs, which include *NST1*, *SND1*, *NST2*, *VND6*, and *VND7*, exert influence over either Tier 2 or Tier 1 TFs [13,23,24].

While there has been considerable research on the process of seasonal wood growth in model plants, the intricate molecular mechanisms behind the initiation and growth of vascular cambium, vascular structuring, and how trees in northern temperate zones respond to environmental stressors are not yet fully understood. Time-course transcriptomics analysis stands out as a pivotal technique to delve into seasonal plant development. This approach facilitates the contrasting study of gene expression at the genomics tier across varied developmental phases. To illustrate, a study by Li et al. [6] has leveraged transcriptomics analysis to decipher the seasonal reconfiguration of xylem in *Pinus radiata*, focusing on trees of different ages. Similarly, Wu et al. [25] have directed their research toward understanding the cyclical evolution of cambial activity and its association with xylem growth in the Chinese fir.

Elm (*Ulmus pumila*), a prominent hardwood species from the broadleaf category, is a deciduous tree under the Ulmaceae family. Originally hailing from central Asia, it now sprawls across continents, finding its roots in regions of Asia, America, and southern Europe [26]. Today, its robust wood is favored for various applications such as crafting furniture, sculpting, and even shipbuilding. Recent studies have pivoted toward understanding the molecular intricacies of elm’s fruit development and its resilience to salt using transcriptome sequencing [26,27,28]. In a preceding investigation, we delved into the growth dynamics of elm by juxtaposing phenotypic and transcriptomic data from cultivars with differing growth rates [29]. Our findings emphasized the significance of the phenylpropanoid route and lignin metabolism in shaping elm branches. Yet, in the context of this premium timber, revered in the northern temperate regions, the processes underpinning seasonal wood development remain uncharted. Building upon our prior explorations on elm, this study embarked on a detailed examination of the anatomical, physiological, and transcriptomic shifts occurring in its vascular tissue across seasons.

Our overarching goal was to illuminate the fundamental biological processes driving seasonal branch growth in elm. The scientific issues we want to uncover in this study mainly include the following: (1) How can elm trees adapt to the environment in the sampling area? (2) What is the mechanism and by which time-point the elm cambium breaks dormancy and begins activity? (3) What is the dynamic mechanism of branch thickening and secondary wall synthesis in elm trees within a year? (4) What is the correlation between sugar content and lignin/cellulose synthesis? Importantly, we must point out the scientific research gap of this study: we did not experimentally verify any of the genes mentioned, marking a constraint in our present research; the sampling genotyping used in the research is relatively single; and the adaptive mechanisms excavated may not be comprehensive.

## 2. Results

### 2.1. The Anatomical Characteristics of Vascular Tissue during the Seasonal Cycle

We detected obvious seasonal changes anatomically in the vascular tissue of elm branches during the seasonal development stages from February to September (Figure 1A–K). In February, the cambium of elm branches was released from dormancy and began to develop. In March, the cambium cells formed a secondary phloem outward and a secondary xylem inward (Figure 1C). Elm branches begin to form secondary xylems (earlywood) with larger pore sizes (Figure 1D). In April, the secondary xylem of the elm tree thickened significantly (Figure 1E). In May, the elm tree began to form a secondary xylem (latewood) with smaller pore sizes and tightly arranged cells (Figure 1F). From June to August, the secondary xylem significantly thickens (Figure 1G–I). From September, the activity of the cambium weakened, and the thickening of the secondary xylem slowed down (Figure 1J,K).

### 2.2. Transcriptome Profiling of Elm

Total mRNA from cells of 18 samples of elm were sequenced using the Illumina system. The pair-end reads obtained from 18 samples of elm are shown in Appendix A. In total, we obtained more than 50 million raw reads for each elm sample, with at least 21 million reads for each condition and time point. More than 76% of HQ reads from individual samples (control and salinity) of elm could be mapped on the elm genome (Appendix A). The assembly of mapped reads resulted in the identification of a total of 46,950 transcripts in elm.

### 2.3. Differentially Expressed Genes (DEGs) Detection

The relative levels of genes’ expression in elm branches during the seasonal cycle (February, March, May, June, August, and September) were evaluated by the fragment per kilobase of exon model per million mapped reads (FPKM) values. When transcripts were compared at each month, the number of DEGs between time points was highest for June vs. February 10,588 (6811/3777) and March vs. February 9795 (6792/3003) and lowest for September vs. August 1476 (934/542) and June vs. May 2693 (1004/1689) (Table 1, Figure 2A).

### 2.4. DEGs at Different Time Points and Function Enrichments

To further provide insights into the functional transitions along branch development in elm, we clustered the 17,440 DEGs into twelve clusters using the Euclidean distance clustering algorithm (Figure 2A, Appendix A). The GO annotation was performed to assign genes to functional categories for each cluster (Figure 2B). Genes belonging to clusters 9 and 11 (C9, C11) were mainly expressed in February, and genes in clusters 4 and 8 (C4 and C8) were synchronously up-regulated from March to May. The February was best represented by 1709/1337 differentially expressed genes in C9/C11 (Figure 2A,B). This cluster contained a set of genes related to “response to cold”, “response to water deprivation”, “response to abscisic acid”, and “regulation of DNA-templated transcription” (Figure 2B). Genes included in cluster 8 (C8) were up-regulated in March and participated in the “carbohydrate metabolic process”, “unidimensional cell growth”, and “cytokinin-activated signaling pathway” (Figure 2B, Appendix A). The genes in C4 (including 1880 genes) were highly expressed in the samples of March/May and represented by genes related to “carbohydrate metabolic process”, “transmembrane transport”, and “plant-type secondary cell wall biogenesis” (Figure 2B, Appendix A). Totals of 3315 genes included in cluster 1 (C1) were up-regulated in June and participated in “translation”, “transcription, DNA-templated”, and “fatty acid metabolic process” (Figure 2B, Appendix A).

### 2.5. KEGG Enrichment Result of DEGs

The top five KEGG pathways enriched by DEGs of clusters are represented in Table 2. The KEGG pathway “Metabolic pathways” (ko01100) and “Biosynthesis of secondary metabolites” (ko01110) were enriched by DEGs of C9–C11 and C4–C8. “Carbon metabolism” (ko01200) was annotated by 57 over-expressed genes in March and May (cluster C4–C8), and 54 up-regulated genes of C1 were annotated in the KEGG pathway “Biosynthesis of amino acids” (ko01230). Moreover, “Fatty acid degradation” (ko00071) was enriched by genes commonly up-regulated in August (cluster C3–C7).

### 2.6. Changes in Soluble Sugar, Cellulose, and Lignin during Elm Branch Development

Notable differences in the soluble sugar, cellulose, and lignin content were observed, and they changed dynamically with the progression of the developmental period (Figure 3A). The sugar content of elm branches showed a significant increase from March to May. From May to August, soluble sugars continued to decrease in elm branches. The trend of changes in the content of cellulose and lignin is similar: it continues to increase from March to June, reaches its highest value in June (23% for Lignin and 45% for cellulose), and then continues to decrease.

### 2.7. The Differentially Expressed TFs

By inferring from the TFs data in *A. thaliana*, we identified a total of 566 TFs differentially expressed in salt stress responding in elm (Appendix A). Totals of 17.3% (98 of 566) and 15.4% (87 of 566) TFs were significantly enriched in DEG clusters C8 (March) and C9 (February). We identified 58, 28, and 35 differentially expressed genes in C1, 2, and 4, which contain key genes for vascular development from March to June (Appendix A).

### 2.8. WGCNA Co-Expression Network Construction

A total of 22,950 genes were included in WGCNA. Then, the soft threshold was determined by scale independence and mean connectivity analysis of modules with different power values ranging from 1 to 20. In our study, power = 10 was set to guarantee high scale independence (near 0.8) and low mean connectivity (near 0) (Appendix A). The mergeCutHeight was set as 0.25, and a total of 30 modules were generated and displayed with different colors (Appendix A). All analyzed genes were included in the 30 modules. We predicted the regulatory relationship of differentially expressed TF in C1, 2, and 4 and plant vascular development key genes by extracting co-expression relationships from the WGCNA network. Totals of 3690 DEG co-expression links were filtered (Appendix A). This network predicted 94 transcription factors for vascular tissue development in elm, such as MYB6, MYB67, VRN1, and WRKY32.

### 2.9. Real-Time Quantitative PCR Validation

To verify the RNA-Seq results, an alternative strategy was selected for the dysregulated unigenes. To verify the RNA-seq results, four over- and four under-expressed unigenes were selected for validation. Primers were designed to span exon–exon junctions (Appendix A). In most cases, the gene expression trends were similar between these two methods. The correlation between the two sets of data was R^2^ = 0.77, and the result is shown in Appendix A. For example, the homolog of phloem filament formation protein, Cluster-11764.16786, which was detected by RNA-Seq as an up-regulated unigene in branch samples of March and May, was also detected significantly overexpressed by the qRT-PCR method (Appendix A).

To verify the expression similarity of genes among three elm trees, 50 unigenes were selected for validation. Primers were designed to span exon–exon junctions (Appendix A). In most cases, the gene expression trends were similar among the three elm trees. The correlation between tree1 and tree2 was cor = 0.692, and the correlation between tree1 and tree3 was cor = 0.911. The correlation test results and expression patterns of seven representative genes in the three tree samples are shown in Appendix A. For example, for the homolog of secondary cell-wall-biosynthesis-participating gene *CESA4*, Cluster-11764.29560, similar expression patterns of this gene were observed in the branch samples from three elm trees: up-regulated expression began in March, reached its highest level in May, and then began to decline (Appendix A).

## 3. Discussion

The process of seasonal wood growth is among the planet’s most vital biological phenomena. Yet, our comprehension of the foundational molecular activities related to vascular cambium onset, vascular structuring, and xylem differentiation remains rudimentary. Lately, comprehensive transcriptomic analyses of wood formation in Populus, Pinus, white teak (*Gmelina arborea* Roxb), *Eucalyptus grandis*, and *Liriodendron chinense* have been brought to light. Siberian elm is used as an ornamental tree/shrub, especially in temperate countries, and offers multiple applications across landscaping. In this research, we undertook a holistic analysis spanning anatomical observations, physiological examinations, and sequential transcriptomic insights to identify key biological sequences that influenced seasonal branch development in elm. This species showcased an innate ability to regulate transcription and manage abiotic stress, particularly adapting to water scarcity and cold conditions during February. Notably, in elm, sugar concentrations peaked in May, while cellulose and lignin content surged to their maximum in June. An inverse relationship was observed between sugar concentrations and the combined lignin/cellulose content during May and June, hinting at a potential carbon redirection towards lignin and cellulose production. Predominant genes engaged in the creation of cellulose and xylan for secondary wall structure exhibited an up-regulation in May and/or June. Our exhaustive transcriptomic study was orchestrated to unearth the molecular intricacies that governed seasonal wood growth in elm.

In the elm’s natural habitat during February, daily temperatures usually hover at lower extremes with sparse rainfall (Figure 3A,B). For trees in temperate zones, winter’s chilly climes are a predominant constraint for their growth and cultivation [30]. In the elm tree, there is a discernible inverse relationship between seasonal average rainfall/temperatures and the number of genes that are activated in response to water scarcity and cold conditions. This highlights the tree’s innate capacity to recalibrate its genetic activities to counter the challenges of drought and cold during winter months. The genes grouped in the differentially expressed gene (DEG) clusters C9 and C11 witnessed a surge in their activities in February (Figure 2A,B). Delving into the functional attributes of these genes, it is evident they played a role in processes like hormonal reactions, regulating gene expression, carbohydrate metabolism, and protein movement, all pointing toward the tree’s adaptive measures against cold stress. Cold-induced stresses can alter the consistency of cell membranes, potentially making them either too rigid or too fluid. These alterations might be detected by membrane-bound proteins, particularly calcium (Ca^2+^) channels and receptor-like kinases (RLKs) [31,32]. In the context of the elm, genes that encode for RLKs, such as *RLK4* (Cluster-11764.20483), *CRK21* (Cluster-11764.20280), and *CRK25* (Cluster-11764.23110), are anticipated to be cold-resilient (Figure 3A,B). In the model plant *A. thaliana*, the CBF/DREB1 pathway stands out as a crucial modulator in response to cold stress. Intriguingly, 14-3-3 proteins can interfere with CBF1/3 proteins, thereby moderating the CBF pathway to ensure that the plant does not overreact to cold conditions [33]. Within the elm’s genetic repertoire in February, there is a heightened activity of DREB1 homologs (Cluster-11764.14897), 14-3-3 protein gene GRF12 (Cluster-11764.26566), and CRPK1 (Cluster-11764.7396). This activity underscored the importance of these genes in signaling and refining responses to cold stress (Figure 3B). When exposed to chilly environments, plants swiftly triggered CBF expression through a cascade of TFs, including *ICE1*, *CCA1*, *HSFA1*, and *CZF1* (Figure 3B). The enhanced expression of these pivotal genes during February suggested that akin to model plants, elm trees also deployed similar genetic mechanisms to confront cold-induced stresses.

The annual rainfall and the number of drought stress gene expressions indicate that drought stress continues to affect the growth and development of elm trees between February and June. The genes of clusters C9 and C11 are involved in transcriptional regulation and plant hormone response, which indicates these processes are related to drought stress response. *ERECTA* genes function as transcriptional regulators and are among the best-characterized genes affecting drought tolerance [34,35]. Six homologs of *ERECTA* were identified in elm; these genes all had the highest expression levels in February (Table 3). The transcription factors of the Asr (abscisic acid, stress, ripening induced) family of genes are plant-specific and stress-regulated components of the ABA-dependent pathway [36,37]. In elm, the homolog of *Asr is* highly expressed in February and June. Furthermore, drought-responsive element-binding (DREB) protein-encoding genes are plant-specific, stress-regulated transcription factors that belong to the AP2/EREBP family and are in the ABA-independent pathway for drought stress response [38]. In this study, three homologs (Cluster-11764.29311, Cluster-11764.8458, and Cluster-11764.22853) of DREB2 were found to be highly expressed in February. These results suggested that these genes in elm possibly played analogous roles in drought stress response.

This research shed light on the seasonal dynamics of the vascular cambium activity in elm, focusing on its anatomical and molecular underpinnings. Notably, at high latitudes, the vascular cambium’s activity is known to be deeply seasonal [3,4]. The anatomical study unveiled that by March, the cambium in elm roused from its dormant state and commenced its developmental journey. This process involved the formation of a secondary phloem externally and a secondary xylem internally (Figure 1C). To delve into the molecular orchestrators of this process in elm, the research sought to determine the seasonal expression patterns of key genes by comparing them to those already documented in model plants (Table 4). A case in point is in Populus, where a surge in temperatures during early spring is believed to trigger the transcription of *CDKB* and *CYCB* homologs in the cambium [12]. This transcriptional activation is pivotal for the division of cambium cells. Mirroring this pattern, the elm also showcased a heightened activity of *CDKB* (Cluster-11764.24087) and *CYCB* homologs (Cluster-11764.6956 and Cluster-11764.30239) in March. This suggested that these genes in elm possibly played analogous roles in steering the division of cambium cells. Auxin, a crucial plant hormone, plays an instrumental role in guiding the initiation and sustenance of procambial stem cells (Weijers and Wagner, 2016). In particular, ARF5 is galvanized in reaction to auxin and facilitates an increase in vascular initial cells. It achieves this by kick-starting the expression of the auxin efflux carrier gene, *PIN1* [9]. In the context of elm, the heightened expression of *ARF5* homologs (Cluster-11764.24231), along with *PIN1* (Cluster-11764.20194) and *PIN3* (Cluster-11764.24634), underscored the pivotal role auxin played in the inception of vascular cambium in this tree species. Lastly, *WOX4* emerged as a vital cog in the machinery regulating vascular cambium division (Table 4). It is involved in the activation of a specialized transcriptional network exclusive to the cambium. The amplified expression of *WOX4* (Cluster-11764.10969) in the months of February and March reinforced its essential role in the development of the vascular cambium in elm.

This investigation underscored the pivotal role of carbohydrates in fueling the growth of wood vascular tissue. The relationship between carbohydrate availability and cell wall component synthesis, specifically cellulose and lignin, has been illuminated by observing trends in plants like Cassava, where starch accumulation exhibited a negative correlation with cellulose and lignin contents [39,40]. In the present study on elm, physiological assessments revealed a notable trend in carbohydrate dynamics: soluble sugars rose from March through May and then waned from May to June (Figure 4A). Meanwhile, the synthesis of cellulose and lignin kept an upward trajectory from March to June (Figure 4A). The implication here is clear. Starting in May, the elm deployed carbohydrates as a carbon resource to synthesize the abundant cellulose and lignin necessary for the formation of the secondary wall. This physiological observation aligned well with the trends captured in the transcriptomic analysis (Figure 4A). Specifically, genes grouped in DEG clusters C8 and C4 displayed a significant up-regulation around March and/or May (Figure 2A,B). A closer look at the functional enrichment analyses of these genes suggested that they were predominantly engaged in processes such as carbohydrate and lipid metabolism, SCW biogenesis in plants, and meristem development. These processes collectively contributed to the development of vascular tissues in elm during the spring season. For wood formation, the relative activities of sucrose synthase (Suc synthase) and invertase in the vascular tissue are instrumental in determining the balance of carbon partitioning between the synthesis of cell wall components and storage [4]. The discovery of three prominently expressed SUC genes in March and/or May (Cluster-11764.17001, Cluster-11764.28335, and Cluster-11764.8128) corroborated the central role of sucrose metabolism during this period (Figure 4B).

A salient feature of cellulose biosynthesis is its production at the cell surface, facilitated by CESAs [13,17,18]. The study identified 13 CESAs in elm that demonstrated elevated expression levels in May. These were found to be homologous to the well-documented secondary wall CESAs in other species, including CESA1, CESA4, CESA5, CESA6, and CESA10 (Figure 4B). This finding established the significance of these CESAs in steering the synthesis of cellulose within the xylem of the elm tree. The synthesis of xylan, a major hemicellulose component in the SCWs of vascular plants, is a complex process. Unlike cellulose, which has a fairly straightforward linear structure, xylan possesses reducing end oligosaccharides and numerous side chains. This structural intricacy requires a diverse array of enzymes to facilitate its biosynthesis. The first steps of xylan biosynthesis involve the conversion of UDP-glucose (UDP-Glc) to UDP-glucuronate, a function attributed to UDP-glucose dehydrogenase (UGD) [41]. Subsequently, the produced UDP-glucuronate is converted to UDP-xylose by UDP-xylose synthase (UXS). UDP-xylose then serves as the building block for the xylan backbone. In this study, elevated expression of the elm homologs of UGD (Cluster-12691.0 and Cluster-11764.28293) and UXS (Cluster-6863.0) during May indicated that these genes likely dictated these crucial initial stages of xylan biosynthesis in the elm tree (Figure 4B). Further synthesis and elongation of the xylosyl backbone were orchestrated by three primary genes: IRX9, IRX10, and IRX14 [19,20]. The dataset from this investigation revealed that the elm homologs of these genes (Cluster-11764.6290 for IRX9, Cluster-11764.22256 for IRX10, and Cluster-11764.10197 for IRX14) showcased pronounced expression in May and/or June (Figure 4B). This finding suggested that these months were pivotal for the extension of the xylan backbone in elm. Additionally, the synthesis of the reducing end oligosaccharides and subsequent modifications to the xylan structure involves various enzymes. Noteworthy among these are the GATL proteins (responsible for the synthesis of the reducing end) and GUX proteins (playing a role in xylan modification). The study mentioned that genes homologous to GATL and GUX in Arabidopsis are delineated in Figure 4B. By piecing together the temporal expression patterns of these genes during spring and early summer, a more holistic picture of xylan biosynthesis in elm emerged. This temporal mapping of gene expression aided in predicting the main enzymatic players at each step of the synthesis, facilitating a deeper understanding of the mechanisms that underscored xylan formation in elm.

Lignin is an intricate phenolic substance that imparts rigidity and water resistance to cell walls, primarily being a mix of p-hydroxylphenyl (H), guaiacyl (G), and syringyl (S) components [15]. The crucial reactions within the overarching phenylpropanoid process are driven by enzymes like phenylalanine ammonia-lyase (PAL), cinnamic acid 4-hydroxylase (C4H), and 4-coumarate-CoA ligase (4CL) [13,15,42]. This research revealed an elevation in the expression of PAL (Cluster-11764.18834), 4CL (Cluster-11764.19115 and Cluster-11764.11333), and C4H (Cluster-11764.3094) genes during May or June (Figure 4C). This observation underscored the relationship between the primary phenylpropanoid route and the lignin formation process in elm branches. In the context of land plants, the enzyme hydroxycinnamoyl-CoA shikimate/quinate hydroxycinnamoyl transferase (HCT) emerges as the key contributor to lignin formation [43]. An analogous gene to HCL (Cluster-11764.26370) exhibited heightened expression in May. Furthermore, during May or June, corresponding genes for key enzymes instrumental in lignin production, including coumarate 3-hydroxylase C3H, caffeate/5-hydroxyferulate 3-O-methyltransferase COMT, cinnamyl alcohol dehydrogenase CAD, and ferulate 5-hydroxylase F5H, exhibited increased expressions (Figure 4C). These data underscored the significance of lignin and its associated metabolic pathways in the development of elm branches during those months.

TFs play a pivotal role in shaping and guiding the growth and structure of wood vasculature [4,22]. Based on TF data from *A. thaliana*, we pinpointed 566 differentially expressed TFs in elm. Breaking down further, 58 TFs were enriched in the DEG cluster C1 (June), 28 in C2 (May), and 35 in C4 (March and May) (Figure 5A). This observation showcased the profound influence of TFs on the seasonal growth dynamics of elm branches. Across the seasonal growth spectrum, several members from the MYB, NAC, HB, C2H2, and bZIP TF families showed differential expressions. Predominantly, TFs influencing the SCW regulatory framework are from either the NAC or MYB TF families [21,22,24]. One of the master regulators, NST1 (Cluster-11764.3959), displayed high expression during March and May (Figure 5A). Both MYB46 (Cluster-4785.0) and MYB103 (Cluster-11764.789), belonging to the Tier2 MYB TFs, recorded increased expression in March and May. The Tier 1 TF BES1, known to manage the transcription of CESA genes [44], showcased a heightened expression in June in its elm counterpart (Cluster-11764.13091). The Tier 1 homeodomain TF KNAT7, which partakes in SCW formation, also had its counterpart up-regulated in May. Collating seasonal expression trends with homologous connections to well-analyzed TFs, we discerned these core TFs pivotal for elm growth (Figure 5A). By assessing the interlinked expression dynamics between genes, we were also able to project several TFs that might be intertwined with established genes vital for wood growth (Figure 5B). The bHLH093 encodes a bHLH TF crucial for the GA-controlled flowering timeline in Arabidopsis. Our research indicated its elm counterpart (Cluster-11764.13485) potentially interacting with cambium development-related genes like REV, ARR7, and PIN1/3 (Figure 5B). MYB proteins represent one of the most expansive TF families in the plant kingdom. In elm, several MYB TF counterpart genes, including MYB3 (Cluster-11764.1809), MYB36 (Cluster-11764.5967), MYB84 (Cluster-11764.27767), MYB305 (Cluster-11764.21578), and CCA1 (Cluster-11764.20181), displayed co-expression with vascular tissue genes, suggesting their potential regulatory roles in branch growth (Figure 5B).

The development of individual xylem cell types has mostly been studied through anatomical observations over the past hundred years; Tung Chia-Chun et al. unraveled the developmental processes of plant xylem through single-cell RNA sequencing (scRNA-seq) analyses. We conducted a correlation analysis between the expression data of elm and the single-cell transcriptome of *P. trichocarpa*; the result is shown in Figure 6. In poplar, genes of libriform fibers (Ptr7) are heavily enriched with SCW biosynthesis genes, especially those for monolignol biosynthesis [45]. Ray parenchyma cells (Ptr8) have moderate expression of monolignol biosynthesis genes, consistent with previous observations that ray parenchyma cells in Picea also contribute to lignification [46]. In elm, the homologs of lignin formation and the primary phenylpropanoid route up-regulated and co-expressed in May, as *PtrHCT1* (*HCT* in *A. thaliana*), *PtrPAL3* (*HCT)*, *PtCesA4 (CESA4)*, *PtrCAld5H1(FAH1),* and *PtrCslA2 (ATCSLA09)* (Figure 6A). These results provided important insights into similarities in developmental trajectories between poplar and Siberian elm. In plants, cell polarization is the specialization of developmental events along one orientation or one direction. In model plants, receptor-like transmembrane kinase I (TMK1) is the key regulator in auxin signaling, and *TMK1*-mediated auxin signaling regulates cell polarity formation and growth of the root of the apical hook [47,48]. In poplar, homologous of *TMK1* are related to vessel elements (Ptr1); in elm, homologous of *TMK1* co-expressed with multiple cell wall development genes (*HCT*, *C3H*, *TBL19*), suggesting the potential regulatory roles of TMK1 in cell wall vessel elements.

Although transcriptome sequencing has the convenience of quickly obtaining a large amount of gene expression information, it still has certain limitations, such as paralog variation. Most plants have undergone multiple rounds of whole genome duplication (WGD) [49], which has long been recognized as an important evolutionary force. It is important to know whether there are correlations between expression divergences of paralogs. Several positive correlations between the divergence of gene and protein expression were identified in mammals. However, a study on sunflower has indicated that there are no correlations [50]. In this study, two unigenes (Cluster-11764.13220 and Cluster-11764.7792) were identified as homologs of leucine-rich receptor-like protein kinase family protein AT1G17230. The expression of these two genes is significantly positively correlated, with high expression in February and no expression detected in the samples from May to September (Table 3). However, there are certain differences in gene expression homologs (Cluster-11764.8337 and Cluster-11764.12867) of leucine-rich repeat transmembrane-type receptor kinase GSO1. Cluster-11764.8337 is expressed in February, while Cluster-11764.12867, although expressed at the highest level in February, is expressed between March and September (Table 3). Functional divergences and differences in the expression of paralogs have arisen due to selective pressures throughout evolution [50,51]. Hence, a comprehensive investigation of the expression patterns of paralogous gene pairs in response to various stresses and a study of correlations between the expression levels and sequence divergences of the paralogs are needed in further analysis.

While we discerned the molecular foundations of bioactive compounds in developing elm branches using anatomical studies, physiological insights, and temporal transcriptome sequencing, we did not experimentally verify any of the genes mentioned, marking a constraint in our present research. Collectively, executing functional studies was imperative to delve deeper into the roles of genes and pathways linked to abiotic stress response and vascular tissue evolution in elm. Advances in genome sequencing technologies combined with efficient trait mapping procedures accelerate the availability of beneficial alleles for breeding and research. Once we have evidence of candidate genes involved in the adaptation or development of elm, the necessary next step to understand the genetic basis of this phenotype is to design in vitro systems, knock-out, knock-down, or knock-in experiments that validate the involvement of such genes in the observed phenotypes [52,53]. Repeats (CRISPR) system alongside the CRISPR-associated protein 9 (Cas9), commonly known as CRISPR/Cas9, which can be used to eliminate, introduce, or replace specific segments of DNA within a targeted site in a genome.

Elm is a tree species with outstanding drought resistance, and there are many elm populations distributed in the Inner Mongolia Autonomous Region in China, which has low annual rainfall in China. Characterizing the population history of elm and identifying loci underlying local adaptation is crucial in functional ecology, evolutionary biology, conservation, and agronomy. Population genomics now provides tools to better integrate selection into a historical framework and take into account selection when reconstructing demographic history [51,54]. Although there is currently no reported whole-genome sequencing data for elm, we believe that we will obtain its whole-genome data in the near future. Transcriptomics provides windows into molecular variation in breeding lines; these windows are closer to phenotype. Complex quantitative trait loci (QTLs) and genome-wide association studies (GWAS) can provide independent sets of markers to complement genetic markers as breeding tools. Even if we do not have the genome of the elm tree, SNP detection by RNA-seq is particularly interesting for non-model model organisms. SNPs detected in expressed regions can be used to characterize variants affecting protein functions and to study cis-regulated genes by analyzing allele-specific expression (ASE) in the interplay of seasonal branch growth and adaptation to abiotic pressures and habitat gradients in elm.

## 4. Materials and Methods

### 4.1. Plant Materials and Anatomical Observation

We chose three robust 15-year-old elm trees for our study; the cultivars used in this study were collected from the Inner Mongolia Autonomous Region in China (39°12′32″ N and 101°38′36″ E). The collected seeds of the cultivar were planted and nurtured in the tree farm of Shandong Normal University in Shandong Province, China, with geographical coordinates 36°32′45″ N and 116°50′2″ E (Figure 1A–C). Monthly samples were taken from the tree from February to September; the sampling date was generally the 15th of each month. For the purpose of our research, we focused on 2-year-old elm branches, which were subject to morphological, physiological, and transcriptomic examinations. In terms of anatomical observations, 10 branch cross-sections gathered from three trees from February to September were preserved in a formalin–alcohol–acetic acid (FAA) solution. Using freehand techniques, slices from these cross-sections were dyed with phloroglucinol. Subsequently, they were examined under a ZEISS Stemi 508 dissecting microscope from Germany, which was complemented with a digital camera for computer-aided visualizations.

### 4.2. Assay of Soluble Sugar, Cellulose, and Lignin Contents

To determine the content of soluble sugars, a 10 g mix branch tissue sample, inclusive of vascular tissue, was dried in an oven set at 65 °C until a consistent weight was achieved. After drying, the tissue was ground to a fine powder using a mortar and pestle and subsequently passed through a 60-mesh sieve. Next, 20 mg of the sieved tissue powder was combined with 5 mL of 80% ethanol, facilitating the extraction of soluble sugars over a span of 24 h at 25 °C. Post extraction, 100 μL of the resultant solution was added to 3 mL of anthrone solution in a 5 mL centrifuge tube. This mixture was then subjected to a boiling water bath for a duration of 15 min. Upon cooling, the absorbance of the reaction solution was assessed at 620 nm. Utilizing a standard curve, soluble sugars were quantified as previously described [39,55].

To analyze the cellulose content, 2 g of the dried, powdered mix branch tissue was accurately measured and placed in a 400 mL flask. To 100 mL of the mixed solution, a solution comprising 50 mL of 2% cetyltrimethylammonium bromide (CTAB, prepared by dissolving 20 g of CTAB in 1 L of sulfuric acid) and 1 mL of decahydronaphthalene was added. This mixture was then boiled for an hour. Following boiling, it underwent suction filtration repeatedly and was cleansed using hot water (90–100 °C) until a neutral pH was attained in the filtrate. Any remaining color was removed from the residue with acetone. The colorless residue was then dried at 65 °C for 12 h, allowed to cool to room temperature within a desiccator, and weighed (noted as W1). Utilizing 72% sulfuric acid, acid-insoluble lignin and other ash content (W2) were separated as per the procedures outlined by Van Soest [56]. The final cellulose content was calculated with the following formula: Percentage cellulose = (W1 − W2)/S × 100%.

A modified Klason technique was employed for assessing lignin content within the branches, as previously detailed by Liu and colleagues in 2018. Samples dried in an oven (0.5 g each) were combined with 5 mL of 72% H_2_SO_4_ and allowed to react at 25 °C for 2 h. This mixture was then diluted using deionized water until it attained a concentration of 3% H_2_SO_4_, after which it was heated to 121 °C for 1 h in an autoclave. Subsequently, the resultant mixture was filtered. After drying at 65 °C for 12 h, the residual matter was used to determine the acid-insoluble lignin content through a gravimetric method [39,57].

### 4.3. RNA Extraction

For transcriptomic evaluations, branches from the months of February, March, May, June, August, and September were chosen. In transcriptome analysis, samples from three trees were treated as three biological replicates. At the time of sampling, from each monthly tissue set, nine small vascular tissue sections were carefully removed from three branches of the three distinct trees and preserved, respectively. For the purposes of RNA extraction and transcriptomic sequencing, we obtained three replicate sets of these vascular tissue sections. For every replicate, a combined weight of 1 g of plant matter was meticulously pulverized under the chilling influence of liquid nitrogen. The resulting material underwent an RNA extraction process using the TRIzol Reagent (sourced from Invitrogen, Carlsbad, CA, USA), adhering strictly to the procedures recommended by the manufacturer. Once extracted, the purity and structural integrity of the RNA were assessed using the RNA Nano 6000 Assay Kit in tandem with the Agilent Bioanalyzer 2100 system (Agilent Technologies, Santa Clara, CA, USA). Additionally, the NanoDrop 2000 spectrophotometer (Thermo Scientific, Wilmington, NC, USA) was utilized for further quality checks.

### 4.4. Illumina Library Construction and Sequencing

The sequencing libraries were crafted using the NEBNext^®^ UltraTM RNA Library Prep Kit for Illumina^®^ (NEB, Ipswich, MA, USA). Unique index codes were integrated to ensure that each sample’s sequences were identifiable. In the initial step, magnetic beads with poly-T oligos were utilized to separate mRNA from the comprehensive RNA. This was then succeeded by the creation of both the first and second strands of cDNA. Thereafter, cDNA fragments that were between 150 and 200 bp in length were cleansed through the AMPure XP system (Beckman Coulter, Beverly, MA, USA). At the conclusion of this process, fragments chosen by size, which also had ligated adaptors, were isolated and enriched via a PCR method. These resultant PCR products were then channeled for extensive sequencing on the Illumina HiSeq X platform (Illumina, San Diego, CA, USA). All the sequenced genetic data from this study were logged into the NCBI Sequence Read Archive (SRA) database (available at https://www.ncbi.nlm.nih.gov/sra). The date of last access was 25 July 2023, with the SRA accession code being PRJNA1007231.

### 4.5. De Novo Assembly of Transcriptome

RNA sequencing and subsequent de novo transcriptome assembly were used to produce reference sequence libraries for mulberry leaves. Each RNA sample was sequenced independently. Subsequently, the cDNA library was developed, and Illumina PE150 sequencing was performed by Novogene Co., Ltd. in Beijing, China. Their official website, accessed on 20 August 2023, is http://www.novogene.com/. To achieve clean reads (i.e., reads containing adapters), those with poly-N and low-quality ones were discarded. The high-quality reads that remained were then employed for transcriptome assembly through the Trinity software version 2.11.0 pipeline using its standard settings [58]. The unigenes assembled de novo underwent BLAST searches and were annotated using various public databases such as NR, NT, Swiss-Prot, Pfam, KOG/COG, Swiss-Prot, KEGG Ortholog database, and Gene Ontology. The threshold set for the E-value during this process was 1 × 10^−5^.

### 4.6. Calculation of Gene Expression in Elm Branches

In this research, 18 separate cDNA libraries from elm samples were developed utilizing the PE150 sequencing method. We determined the number of reads associated with each gene using featureCounts v1.5.0-p3. Then, the gene’s length and the read count linked to that particular gene were employed to figure out each gene’s FPKM. The DESeq2 R package (version 1.20.0) was used to analyze differential gene expression between salt-exposed samples and their controls at varying time intervals. To regulate the false discovery rate, *p*-values were adjusted using the method proposed by Benjamini and Hochberg. DEGs were identified with criteria of a *p*-value below 0.05 and an absolute log2 fold change greater than 1.

### 4.7. Gene Function Annotation

The unigenes of *U. pumila* were mapped to *A. thaliana* and *P. trichocarpa* gene IDs by sequence similarity searching against the genome of *A. thaliana* and *P. trichocarpa* with an E-value cutoff of 1 × 10^−5^. *A. thaliana*’s TF information was sourced from plantTFDB (http://planttfdb.cbi.pku.edu.cn/, accessed on 15 May 2023). Elm’s TFs were identified by comparing them to those of *A. thaliana*. Research conducted by Wang [4], Agustí [59], and Kumar [13] has provided pivotal genes for plant vascular tissue evolution, which are categorized into three operations: cambium development, plant vascular development, and SCW biosynthesis.

### 4.8. GO and KEGG Enrichment

We employed the topGO toolkit within R for GO enrichment analysis of DEGs and various module genes. The DEGs of elm unigene IDs were transferred to the Arabidopsis TAIR locus IDs during the MapMan analysis. The software KOBAS version 2.1.1 was used to test the statistical enrichment of differential expression genes in KEGG pathways in elm.

### 4.9. WGCNA Co-Expression Network Construction

To build the WGCNA co-expression network, the FPKM gene values were adjusted and normalized. Genes displaying minimal expression variance across samples were excluded. Using the WGCNA toolkit in R (Version 3.3.2) [60], a co-expression grid was formed for the remaining genes. Following the arrangement of samples, the soft threshold for module assessment was defined through the scale autonomy and average link analysis of modules at varying power levels. This power ranged from 1 to 20, with subsequent calculations for scale freedom and mean connectivity. The power was determined when the scale freedom reached 0.9. To categorize similar gene expression patterns into distinct modules, the mean distance (with a least threshold of 30) and a merging height of 0.25 were applied to construct a hierarchical cluster tree of the TOM matrix. Additionally, a module cluster tree and a principal gene proximity heatmap among modules were produced. The visualization of these co-expression grids was facilitated using Cytoscape 2.8.2.

### 4.10. Correlation Analysis of Xylem Cell Development between Elm and P. trichocarpa

By integrating scRNA-seq-lcmRNA-seq correlation, scUPlcmUP gene distribution, and known gene functions, Tung Chia-Chun et al. revealed the single-cell clusters of vessel elements (Ptr1), libriform fibers (Ptr7), and ray parenchyma cells (Ptr8) in *P. trichocarpa* [45]. Based on the homologous relationship between poplar and elm genes, we first identified the homologous genes of the genes in the three clusters (Ptr7, Ptr1, Ptr8) in elm and further screened out the genes potentially involved in the above three processes in elm through the expression patterns in May and June. By assessing the interlinked co-expression relations between genes, we were also able to project several potential genes that might be intertwined with established genes vital for xylem cell development.

### 4.11. qRT-PCR Verification

RNA-seq analysis results were validated using qRT-PCR. RNA samples were treated with DNaseI to digest any DNA and then converted to cDNA using the PrimeScript RT Reagent Kit with gDNA Eraser (Takara, Dalian, China). For the qRT-PCR test, eight randomly selected DEGs were used (namely, Cluster-11764.17001 through Cluster-11764.27024). The reference gene used was the ortholog (Cluster-11764.18535) of *A. thaliana*’s alpha unit of the elongation factor-1 complex found in mulberry. Using Premier 5.0 software, gene-specific primers, each 19–22 base pairs long, were devised (Appendix A). To verify the consistency of gene expression in samples collected from three trees, we selected 50 genes for RT PCR experiments. These genes are involved in processes such as “translation elongation”, “mitosis”, “transcriptional regulation”, “cell wall synthesis”, “sucrose synthesis”, and “lipid synthesis”, as well as 13 randomly selected genes. The primers of these genes are shown in Appendix A. The qRT-PCR procedure was executed using the ABI7500 Real-Time PCR System (ABI, New York, NY, USA) and the SYBR Green qPCR Master Mix (DBI, Rheine, Germany). All tests were carried out three times, and a melting curve analysis was used to assess the specificity of amplification. The 2^−(ΔΔCt)^ technique was used to compute the relative gene expression levels. Pearson’s correlation test is used to test the similarity between two sets of data.

## 5. Conclusions

This study delved into the anatomical, physiological, and transcriptomic variations in elm’s vascular tissue throughout the annual cycle. The most significant differences in DEGs between specific months were observed in the comparison between June and February with 10,588 DEGs (6811 up-regulated/3777 down-regulated) and March and February with 9795 DEGs (6792 up-regulated/3003 down-regulated). Elm enhanced its transcriptional responses and resilience to abiotic stress, particularly addressing challenges like moisture scarcity and low temperatures in February. The peak sugar concentration in elm was noted in May, while the apex for cellulose and lignin was in June. The inverse relationship between sugar levels and the content of lignin/cellulose from May to June suggested a potential shift in carbon allocation towards lignin and cellulose production. Key genes driving the production of cellulose and xylan, essential for secondary wall development, exhibited elevated expressions in either or both May and June. A comprehensive analysis of the transcriptome was conducted to discern the molecular intricacies governing seasonal wood production in elm.

## Figures and Tables

**Figure 1 ijms-24-14976-f001:**
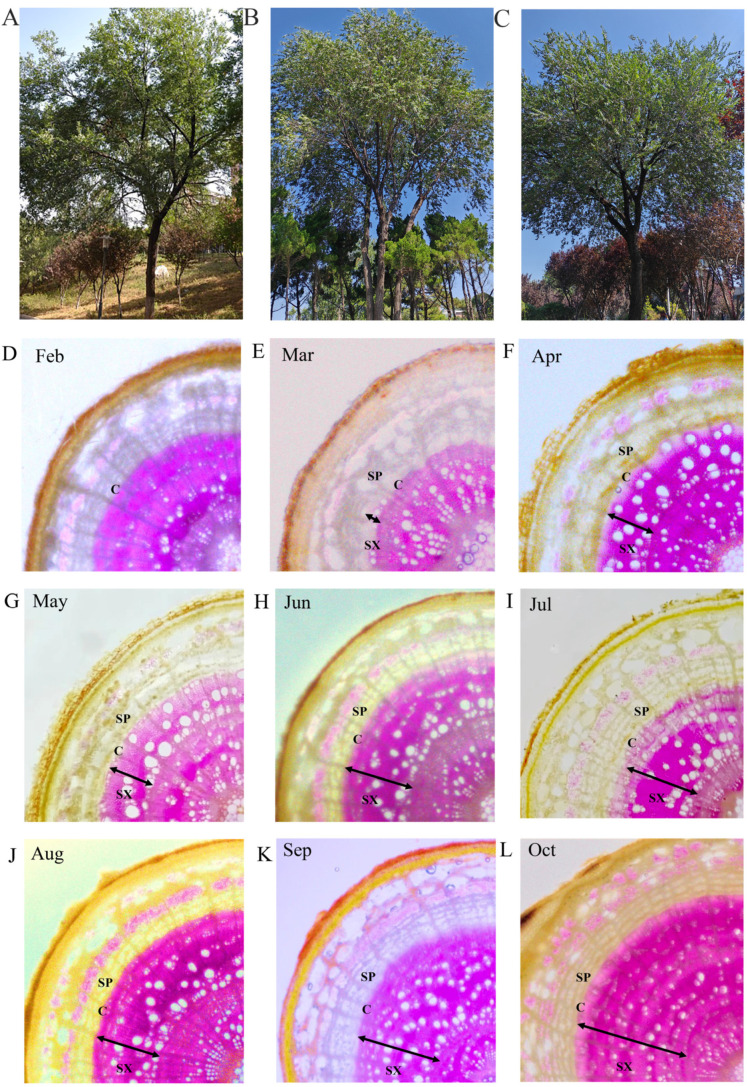
(**A**–**C**) Three elm trees used for analysis. (**D**–**K**) Seasonal anatomical changes in the vascular tissue of elm: (**D**) February; (**E**) March; (**F**) April; (**G**) May; (**H**) June; (**I**) July; (**J**) August; (**K**) September; (**L**) October. C, cambium region; SP, secondary phloem; SX, secondary xylem.

**Figure 2 ijms-24-14976-f002:**
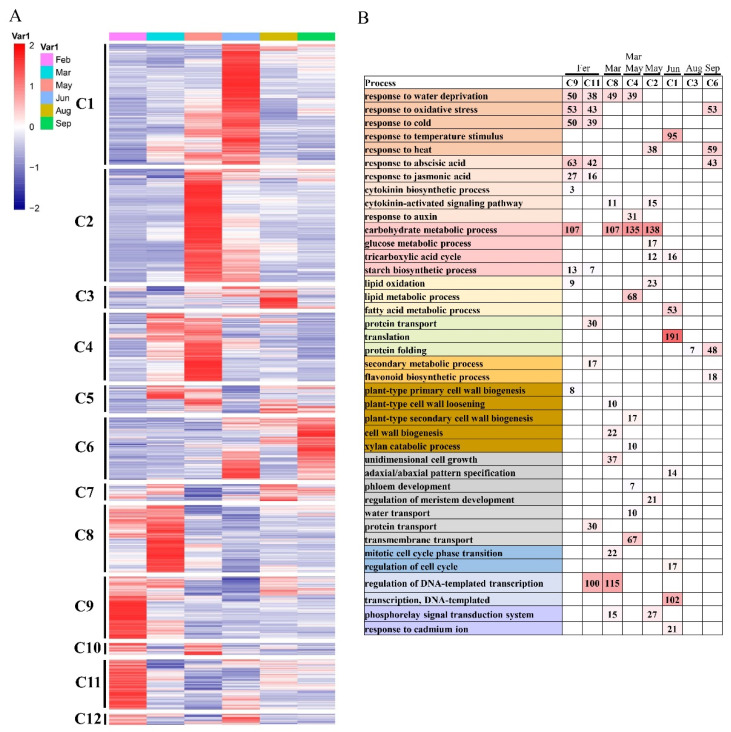
(**A**) Gene expression pattern and functional transition of fast- and slow-growing genotypes. Expression patterns of 17,440 differentially expressed genes (DEGs) in twelve DEG clusters. (**B**) Gene ontology (GO) enriched in twelve DEG clusters. Representative significant categories (*p*-value < 0.05) are displayed.

**Figure 3 ijms-24-14976-f003:**
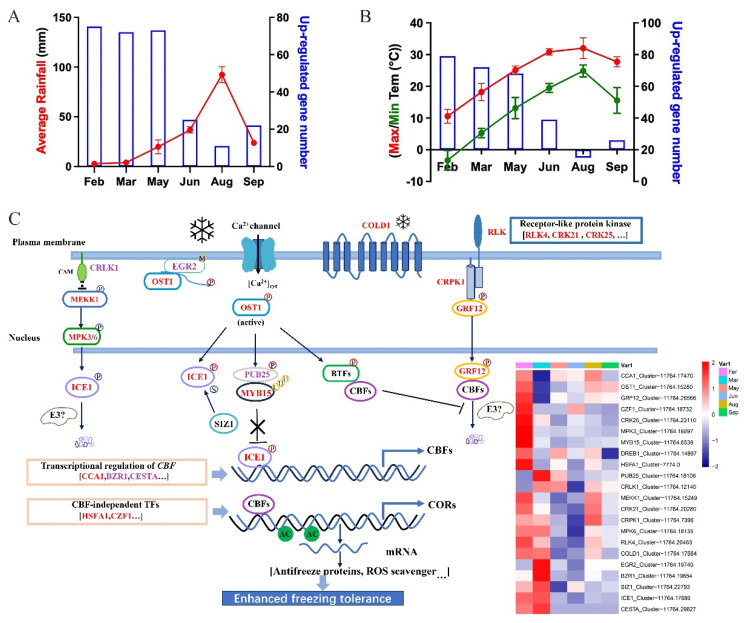
(**A**) Gene number of “response to water deprivation” and average rainfalls. In each plot, blue bars (right *Y*-axis) indicate gene number of “response to water deprivation” found in elm branches sampling in different months, while red line (left *Y*-axis) indicates seasonal average rainfalls in sampling location. (**B**) Gene number of “response to cold” and max/min temperature. In each plot, blue bars (right *Y*-axis) indicate gene number of “response to cold” found in elm branches sampling in different months, while red/green line (left *Y*-axis) indicates max/min temperature in sampling location. (**C**) Predicted regulatory mechanisms of cold response in elm. Cold signaling involves transcriptional, post-transcriptional, and post-translational events. The genes up-regulated in May/June were marked with red/purple color, and others were marked by black. The expression levels (FPKM) of selected differentially expressed genes in May and/or June are shown on the right. A red color indicates that the gene is highly expressed in the branch samples.

**Figure 4 ijms-24-14976-f004:**
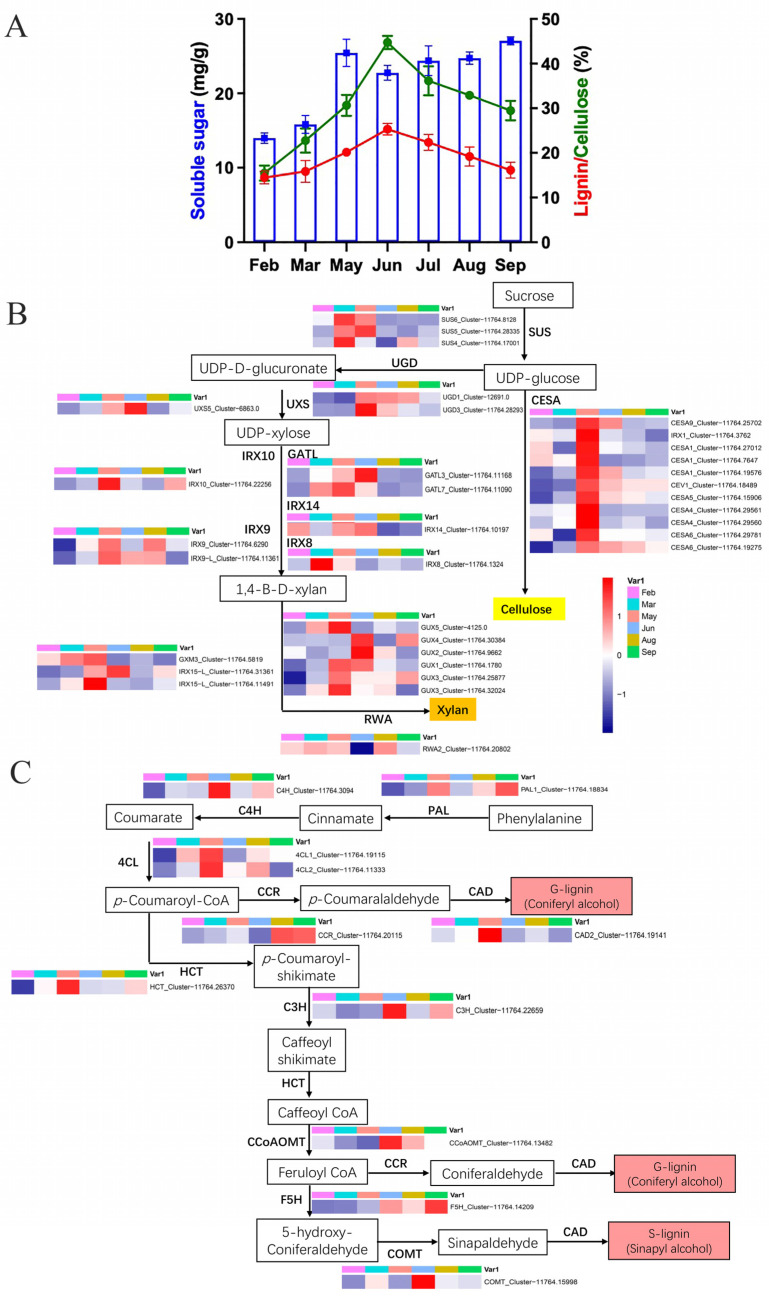
(**A**) Soluble sugar, cellulose, and lignin contents. Blue bars (right *Y*-axis) indicate soluble sugar content, while red/green lines (left *Y*-axis) indicate cellulose/lignin contents. (**B**) Identification of major players within cellulose and xylan biosynthetic pathways for secondary wall. Metabolites in each step of the biosynthetic pathway are shown in the box, and all related genes are shown to the side. Color gradient according to the expression levels (FPKM) was visualized. (**C**) Identification of major players within monolignol biosynthetic pathways for secondary wall.

**Figure 5 ijms-24-14976-f005:**
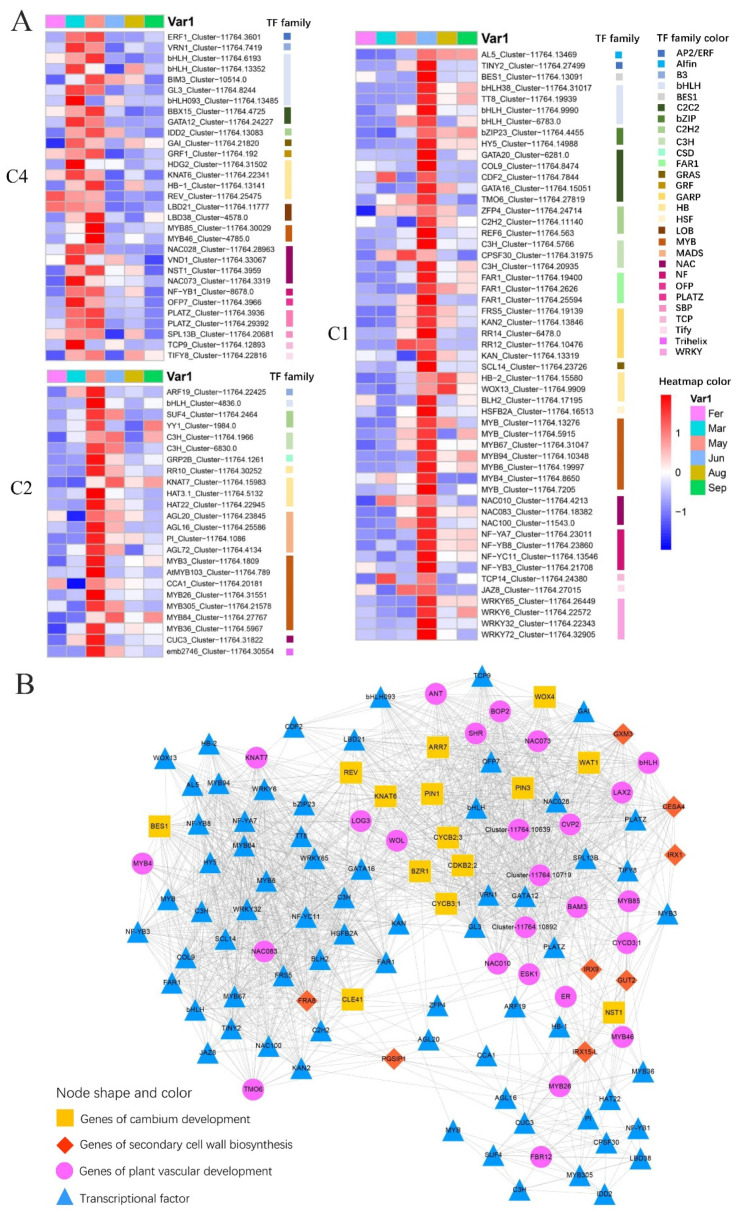
(**A**) Expression pattern of TFs, including in DEG clusters C4, C2, and C1, in six time point samples (February, March, May, June, August, and September). (**B**) The predicted regulatory relationship of differentially expressed TF in C1, 2, and 4 and plant vascular development key genes by extracting co-expression relationships from WGCNA network. The genes of cambium development/secondary cell wall biosynthesis/plant vascular development were represented by yellow square/red diamond/pink origin. The predicted transcription factors are represented by blue triangles.

**Figure 6 ijms-24-14976-f006:**
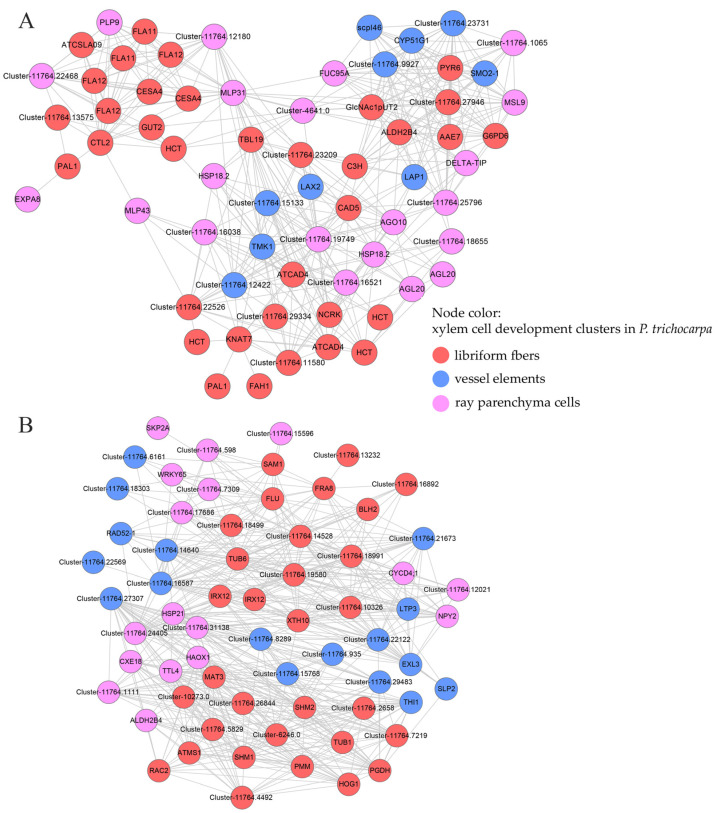
The predicted regulatory relationship of homologs of xylem development gene in *P. trichocarpa* by extracting co-expression relationships from WGCNA network. (**A**) genes up-regulated in May in elm; (**B**) genes up-regulated in June in elm. The homolog genes of libriform fibers (Ptr7)/vessel elements (Ptr1)/ray parenchyma cells (Ptr8) were represented by red/blue/pink.

**Table 1 ijms-24-14976-t001:** Number of differentially expressed genes between different months.

Month	Feb	Mar	May	Jun	Aug
Feb	-				
Mar	6057 (4003/2054)	-			
May	9795 (6792/3003)	4659 (2913/1746)	-		
Jun	10,588 (6811/3777)	8833 (4179/4654)	2693 (1004/1689)	-	
Aug	5193 (3019/2174)	4860 (2244/2616)	3315 (748/2567)	4720 (1601/3119)	-
Sep	6386 (3961/2425)	6779 (2925/3854)	4934 (1360/3574)	3298 (1507/1791)	1476 (934/542)

**Table 2 ijms-24-14976-t002:** The top KEGG pathways enriched by DEGs.

Cluster	ID	Term	Count	*p*-Value
C9–C11	ko01100	Metabolic pathways	355	3.26 × 10^−9^
	ko00195	Photosynthesis	22	1.6 × 10^−4^
	ko00860	Porphyrin metabolism	17	2.71 × 10^−4^
	ko01200	Carbon metabolism	51	8.97 × 10^−4^
	ko01110	Biosynthesis of secondary metabolites	184	1.72 × 10^−3^
C4–C8	ko01100	Metabolic pathways	380	4.77 × 10^−12^
	ko01110	Biosynthesis of secondary metabolites	212	4.7 × 10^−7^
	ko03050	Proteasome	21	2.25 × 10^−5^
	ko01200	Carbon metabolism	57	5.71 × 10^−5^
	ko00250	Alanine, aspartate and glutamate metabolism	18	7.3 × 10^−5^
C2	ko01100	Metabolic pathways	374	5.74 × 10^−17^
	ko01230	Biosynthesis of amino acids	68	1.89 × 10^−12^
	ko01240	Biosynthesis of cofactors	64	3.51 × 10^−11^
	ko01110	Biosynthesis of secondary metabolites	215	1.27 × 10^−10^
	ko01200	Carbon metabolism	65	8.24 × 10^−9^
C1	ko03010	Ribosome	92	5.9 × 10^−13^
	ko01200	Carbon metabolism	60	3.74 × 10^−6^
	ko01230	Biosynthesis of amino acids	54	1.02 × 10^−5^
	ko00630	Glyoxylate and dicarboxylate metabolism	23	8.99 × 10^−5^
	ko00250	Alanine, aspartate and glutamate metabolism	16	7.23 × 10^−4^
C3–C7	ko03010	Ribosome	66	9.72 × 10^−25^
	ko00280	Valine, leucine and isoleucine degradation	10	3.78 × 10^−4^
	ko00071	Fatty acid degradation	8	4.13 × 10^−3^
	ko00310	Lysine degradation	5	4.56 × 10^−2^
C6	ko01110	Biosynthesis of secondary metabolites	131	1.03 × 10^−13^
	ko00280	Valine, leucine and isoleucine degradation	13	2.4 × 10^−5^
	ko00770	Pantothenate and CoA biosynthesis	10	8.13 × 10^−5^
	ko01100	Metabolic pathways	170	8.32 × 10^−5^
	ko00941	Flavonoid biosynthesis	8	3.52 × 10^−4^

**Table 3 ijms-24-14976-t003:** Expression patterns of genes participating in the drought stress response during different months.

Gene Family	Gene ID	Feb	Mar	May	Jun	Aug	Sep
ERECTA	Cluster-11764.13220	**2.730**	1.453	2.443	0.583	2.693	0.900
	Cluster-11764.29309	**2.790**	0.147	0.010	0.000	0.000	0.000
	Cluster-11764.7792	**3.493**	0.283	0.000	0.000	0.000	0.000
	Cluster-11764.7410	**4.160**	0.660	0.870	0.530	0.323	0.417
	Cluster-11764.8337	**1.957**	1.143	0.033	0.023	0.000	0.020
	Cluster-11764.12867	**11.733**	3.180	5.987	7.133	8.587	4.473
ASR	Cluster-11764.14920	33.927	5.770	4.520	**36.980**	3.447	9.517
DREB2	Cluster-11764.29311	**41.210**	27.883	5.987	13.410	5.787	6.553
	Cluster-11764.8458	**23.427**	7.093	1.243	7.247	3.340	3.710
	Cluster-11764.22853	**134.377**	98.333	50.853	97.947	55.550	41.803
	Cluster-11764.16708	432.907	268.647	171.647	**522.343**	168.580	159.453

Note: The maximum value of gene expression in each month is marked in bold font.

**Table 4 ijms-24-14976-t004:** Expression patterns of developmental functional genes in the cambium during different months.

Gene ID	Gene Name	Feb	Mar	May	Jun	Aug	Sep
Cluster-11764.6956	*CYCB3;1*	1.67	**3.29**	1.48	0.18	1.49	0.59
Cluster-11764.30239	*CYCB2;3*	4.56	**16.29**	4.92	0.23	2.26	1.39
Cluster-11764.24087	*CDKB2;2*	11.49	**38.02**	13.43	2.78	7.59	9.06
Cluster-11764.10969	*WOX4*	22.10	**24.00**	19.32	12.91	12.30	11.01
Cluster-11764.12482	*ARR7*	4.23	**12.54**	5.73	1.97	7.25	4.70
Cluster-11764.19654	*BZR1*	66.07	**114.36**	46.52	39.63	60.94	49.88
Cluster-11764.22341	*KNAT6*	12.46	**17.86**	13.95	4.75	9.70	5.65
Cluster-11764.24231	*ARF5*	**21.403**	12.757	15.217	15.197	18.76	19.163
Cluster-11764.24634	*PIN3*	6.00	**21.86**	14.28	11.12	9.29	8.51
Cluster-11764.20194	*PIN1*	23.71	**54.47**	42.99	10.18	20.69	13.43
Cluster-11764.16457	*WAT1*	103.02	201.10	**207.68**	48.20	98.85	74.71
Cluster-11764.25475	*REV*	**35.25**	28.53	26.20	4.60	8.37	5.74
Cluster-11764.6227	*HK3*	0.11	0.22	**3.39**	0.60	0.03	1.06

Note: The maximum value of gene expression in each month is marked in bold font.

## Data Availability

All genetic data have been submitted to the NCBI Sequence Read Archive (SRA) database (https://submit.ncbi.nlm.nih.gov/subs/sra, accessed on 10 August 2023), PRJNA1007231 for *U. pumila*.

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
