# Peer review of "Comprehensive Time-Course Transcriptome Reveals the Crucial Biological Pathways Involved in the Seasonal Branch Growth in Siberian Elm (Ulmus pumila)"

_ijms, 2023, doi:10.3390/ijms241914976_

Round 1

Reviewer 1 Report

The manuscript is well written and well organized.

can be improved

Author Response

Responses to Reviewer 1’s comments:

This manuscript presents the anatomical, physiological, and transcriptomic changes in elm’s vascular tissue throughout the annual cycle. I think the experiments is well designed and the results will be useful for many researchers in the field for future reference. Favorable opinion for publication.

We sincerely thank the reviewer for constructive criticisms and valuable suggestions. For improving the quality of the manuscript, we have made the following modifications to the manuscript:

For figures, we modified Figure1: added new panel B-C to show the other two trees used for analysis. We chose three robust 15-year-old elm trees as three reduplicates for our study. We added Figure 6 to show the predicted regulatory relationship of homologs of xylem development gene in P. trichocarpa by extracting co-expression relationships from WGCNA network. We added Figure S3 to show the the consistency of gene expression in samples collected from three trees, 50 genes were selected for RT-PCR experiments in this analysis.

For tables, we added new Table 2 to show the top KEGG pathways enriched by differentially expressed genes (DEGs). Besides, we added new Table 3 to show expression patterns of genes participating in the drought stress responding.

For manuscript, we rewrote the abstract to focus on key findings and report the sampling genotyping details. We added KEGG enrichment result of DEGs to result section. We added three paragraphs to the discussion section: (1) potential effects of some gene families (ASR, DREB2 and ERECTA) that have previously been linked with abiotic stresses tolerance; (2) the correlation analysis between the expression data of elm and single-cell transcriptome of in P. trichocarpa; (3) limitations of transcriptome analysis, as paralogous variation; (4) prediction of the necessary next step to understand the genetic basis of phenotype in elm, as CRISPR system and population genomics analysis. The sampling genotyping details of three elm trees, and KEGG enrichment analysis were added to the method section.

Comments:

1 Minor comment,

Abstract need to rewrite to focus on key findings.

Answer: we rewrote the abstract to focus on key findings.

Line 11-32

“Timber, the most prevalent organic material on this planet, is the result of a secondary xylem emerging from vascular cambium. Yet, the intricate processes governing its seasonal generation are largely a mystery. To better understand the cyclic growth of vascular tissues in elm, we undertook an extensive study examining the anatomy, physiology, and genetic expressions in Ulmus pumila. We chose three robust 15-year-old elm trees for our study, the cultivar used in this study were collected from the Inner Mongolia Autonomous Region in China and nurtured in the tree farm of Shandong Normal University. Monthly samples of 2-year-old elm branches were taken from the tree from February to September. Marked seasonal shifts in elm branch vascular tissues were observed by phenotypic observation: in February, the cambium of the branch emerged from dormancy, spurring growth; by May, elms began generating secondary xylem, or latewood, recognized by its tiny pores and dense cell structure; from June to August, there was a marked increase in the thickness of the secondary xylem. Transcriptome sequencing provides a potential molecular mechanism for the thickening of elm branches and their response to stress. In February, the tree enhanced its genetic responses to cold and drought stress. The amplified expression of CDKB, CYCB, WOX4 and ARF5 in the months of February and March reinforced their essential role in the development of the vascular cambium in elm. Starting in May, the elm deployed carbohydrates as a carbon resource to synthesize the abundant cellulose and lignin necessary for the formation of the secondary wall. Major genes participating in cellulose (SUC and CESAs honmolgs), xylan (UGD, UXS, IRX9, IRX10 and IRX14) and lignin (PAL, C4H, 4CL, HCT, C3H, COMT and CAD) biosynthetic pathways for secondary wall formation up-regulated at May or/and June. In conclusion, our findings provided a foundation for an in-depth exploration of the molecular processes dictating the seasonal growth of elm timber”

Reviewer 2 Report

The work by Zhang et al employed transcriptomics in a 15-year-old elm tree in order to unveil the functional genetic basis of seasonal branch growth. Even though this is well-designed study that pushes forward the transcriptomics and molecular physiology understanding of a key morphological/adaptive trait in tree species, some amendments are required in order to improve the technical soundness. Specifically, authors are invited to explicitly report the sampling genotyping details in the abstract (L15), enlist specific research hypotheses and be more careful at clearly drawing the line at the Introduction (last paragraph, L119 where the goal is stated) on what the scientific research gap that motivated this report is.

Besides these suggestions to aim for a more hypothesis-oriented argumentation, from a more technical point of view I am afraid the transcriptomic sampling applied in this study may have low power to detect rare functional expression since only one (15-year-old) elm tree was consider, without any genotypic variation/replication whatsoever. Authors need to acknowledge this caveat, and discuss whether a potential lack of power may tend ignoring rare expression profiles. Besides, is there certainty that inferred expression profiles are not contaminated with paralogous variation? I encourage authors to comment on these analyses and discuss this in the manuscript, specially at the end of the discussion section (L417) as a section devoted to potential caveat, and another one envisioning future studies and recommendations capable to amend and bridge these limitations.

I also could not help noticing that authors try performing a metabolic network/pathway synthesis if figure 4. Please complement this with a pathways-enriched analysis as carried out in Agronomy2023 13:1396 (details in the second paragraph of section 2.9, page 7).

On the other hand, authors refer in several occasions to transcriptional responses related to abiotic stress  (in L226 L410 L551). Are there any particular hints on some gene families that have previously been linked with abiotic stresses tolerance? For instance, refer to: (i) Plant Science 2016 242:250 for the ERECTA gene family in association with the AP2 domain, (ii) BMC Genetics 2012 13:58 for the ASR family f in association with the ABA-dependent MYB, and (iii) Theor Appl Genet 2012 125(5):1069-85 for the DREB transcription factor family pleotropic in several pathways with the WRKY transcription factor, all of the above revisited at Genes 2021 12(4):556. I recommend authors to discuss and refer to these cases. Please revisit this point during the discussion section. Other target candidate genes worth exploring and referring to as part of the discussion on abiotic stress tolerance are enlisted in Front Plant Sci 2018 9:128 and Front Genet 2019 10:954.

Last but not least, please also embrace in the novel perspectives section at L417 how could the identified expression patterns source the understanding of seasonal branch growth in other tree species besides elm? Specifically, a major question that authors should prospect in their discussion is how to unlock and effectively utilize these novel gene functionalities. Speed breeding, gene editing, recurrent backcrossing and inter-specific schemes (condensed, contrasted and discussed in Trends Genet 2021 37:1124-1136, and Front Plant Sci 2020 11:583323, please refer to) may offer some insights. Another broader reflection to embrace in this novel perspective section is the interplay of seasonal branch growth and adaptation to abiotic pressures and habitat gradients in an eco-physiological context (refer to the seminal reviews Front Genet 2020 11:564515 and Front Genet 2022 13:910386).

Author Response

Responses to Reviewer 2’s comments:

The work by Zhang et al employed transcriptomics in a 15-year-old elm tree in order to unveil the functional genetic basis of seasonal branch growth. Even though this is well-designed study that pushes forward the transcriptomics and molecular physiology understanding of a key morphological/adaptive trait in tree species, some amendments are required in order to improve the technical soundness.

We sincerely thank the reviewer for constructive criticisms and valuable suggestions. For improving the quality of the manuscript, we have made the following modifications to the manuscript:

For figures, we modified Figure1: added new panel B-C to show the other two trees used for analysis. We chose three robust 15-year-old elm trees as three reduplicates for our study. We added Figure 6 to show the predicted regulatory relationship of homologs of xylem development gene in P. trichocarpa by extracting co-expression relationships from WGCNA network. We added Figure S3 to show the the consistency of gene expression in samples collected from three trees, 50 genes were selected for RT-PCR experiments in this analysis.

For tables, we added new Table 2 to show the top KEGG pathways enriched by differentially expressed genes (DEGs). Besides, we added new Table 3 to show expression patterns of genes participating in the drought stress responding.

For manuscript, we rewrote the abstract to focus on key findings and report the sampling genotyping details. We added KEGG enrichment result of DEGs to result section. We added three paragraphs to the discussion section: (1) potential effects of some gene families (ASR, DREB2 and ERECTA) that have previously been linked with abiotic stresses tolerance; (2) the correlation analysis between the expression data of elm and single-cell transcriptome of in P. trichocarpa; (3) limitations of transcriptome analysis, as paralogous variation; (4) prediction of the necessary next step to understand the genetic basis of phenotype in elm, as CRISPR system and population genomics analysis. The sampling genotyping details of three elm trees, and KEGG enrichment analysis were added to the method section.

Comments:

  1. Specifically, authors are invited to explicitly report the sampling genotyping details in the abstract (L15), enlist specific research hypotheses and be more careful at clearly drawing the line at the Introduction (last paragraph, L119 where the goal is stated) on what the scientific research gap that motivated this report is.

Answer: We sincerely thank the reviewer for constructive criticisms and valuable suggestions. We rewrote the abstract to explicitly report the sampling genotyping details, and added the research hypotheses and the scientific research gap in the introduction section. Please find:

Line 11-32

“Abstract: Timber, the most prevalent organic material on this planet, is the result of a secondary xylem emerging from vascular cambium. Yet, the intricate processes governing its seasonal generation are largely a mystery. To better understand the cyclic growth of vascular tissues in elm, we undertook an extensive study examining the anatomy, physiology, and genetic expressions in Ulmus pumila. We chose three robust 15-year-old elm trees for our study, the cultivar used in this study were collected from the Inner Mongolia Autonomous Region in China and nurtured in the tree farm of Shandong Normal University. Monthly samples of 2-year-old elm branches were taken from the tree from February to September. Marked seasonal shifts in elm branch vascular tissues were observed by phenotypic observation: in February, the cambium of the branch emerged from dormancy, spurring growth; by May, elms began generating secondary xylem, or latewood, recognized by its tiny pores and dense cell structure; from June to August, there was a marked increase in the thickness of the secondary xylem. Transcriptome sequencing provides a potential molecular mechanism for the thickening of elm branches and their response to stress. In February, the tree enhanced its genetic responses to cold and drought stress. The amplified expression of CDKB, CYCB, WOX4 and ARF5 in the months of February and March reinforced their essential role in the development of the vascular cambium in elm. Starting in May, the elm deployed carbohydrates as a carbon resource to synthesize the abundant cellulose and lignin necessary for the formation of the secondary wall. Major genes participating in cellulose (SUC and CESAs honmolgs), xylan (UGD, UXS, IRX9, IRX10 and IRX14) and lignin (PAL, C4H, 4CL, HCT, C3H, COMT and CAD) biosynthetic pathways for secondary wall formation up-regulated at May or/and June. In conclusion, our findings provided a foundation for an in-depth exploration of the molecular processes dictating the seasonal growth of elm timber.”

Line 120-129

“Our overarching goal was to illuminate the fundamental biological processes driving seasonal branch growth in elm. The scientific issues we want to uncover in this study mainly include: (1) how can elm trees adapt to the environment in the sampling area? (2) What is mechanism and by which time-point the elm cambium breaks dormancy and be-gins activity? (3) What is the dynamic mechanism of branch thickening and secondary wall synthesis in elm trees within a year? (4) What is the correlation between sugar content and lignin/cellulose synthesis? Importantly, we must point out the scientific research gap of this study: we did not experimentally verify any of the genes mentioned, marking a constraint in our present research; the sampling genotyping used in the research is relatively single, and the adaptive mechanisms excavated may not be comprehensive.”

  1. Besides these suggestions to aim for a more hypothesis-oriented argumentation, from a more technical point of view I am afraid the transcriptomic sampling applied in this study may have low power to detect rare functional expression since only one (15-year-old) elm tree was consider, without any genotypic variation/replication whatsoever. Authors need to acknowledge this caveat, and discuss whether a potential lack of power may tend ignoring rare expression profiles.

Answer: We sincerely thank the reviewer for constructive criticisms and valuable suggestions. We sincerely apologize for the mistake we made in writing the sampling method. We chose three robust 15-year-old elm trees as three biological reduplicates for our study. We modified the sampling in the method section, added new panel B-C of Figure 1 to show the other two trees (tree2 and tree3) used for analysis. Besides, to show the consistency of gene expression in samples collected from three biological reduplicates, 50 genes were selected for RT-PCR experiments in this analysis. In most cases, the gene expression trends were similar among three elm trees, the correlation between the tree1 and tree2 was cor = 0.692, the correlation between the tree1 and tree3 was cor = 0.911. The correlation test results and expression patterns of 7 representative genes in the three tree samples are shown in new Figure S3.

Line 550-563

“We chose three robust 15-year-old elm trees for our study, the cultivar used in this study were collected from the Inner Mongolia Autonomous Region in China (39°12′32″ N and 101°38′36″ E). The collected seeds of the cultivar were planted and nurtured in the tree farm of Shandong Normal University in Shandong Province, China, with geographical coordinates 36°32′45″ N and 116°50′2″ E (Figure 1A-C). Monthly samples were taken from the tree from February to September, the sampling date is generally the 15th of each month. For the purpose of our research, we focused on 2-year-old elm branches, which were subject to morphological, physiological, and transcriptomic examinations.”

Line 597-598

“In transcriptome analysis, samples from three trees were treated as three biological replicates.”

Line 693-697

“To verify the consistency of gene expression in samples collected from three trees, we se-lected 50 genes for RT-PCR experiments. These genes are involved in processes such as “translation elongation”, “mitosis”, “transcriptional regulation”, “cell wall synthesis”, “sucrose synthesis”, and “lipid synthesis”, as well as 13 randomly selected genes.”

Line 229-238

“To verify the expression similarity of genes among three elm trees, 50 unigenes were selected for validation. Primers were designed to span exon–exon junctions (Table S7). In most cases, the gene expression trends were similar among three elm trees, the correlation between the tree1 and tree2 was cor = 0.692, the correlation between the tree1 and tree3 was cor = 0.911. The correlation test results and expression patterns of 7 representative genes in the three tree samples are shown in Figure S3. For example, the homolog of Secondary cell wall biosynthesis participating gene CESA4, Cluster-11764.29560, similar expression patterns of this gene were observed in the branch samples from three elm trees: upregulat-ed expression began in March, reached its highest level in May, and then began to decline (Figure S3).”

  1. Besides, is there certainty that inferred expression profiles are not contaminated with paralogous variation? I encourage authors to comment on these analyses and discuss this in the manuscript, specially at the end of the discussion section (L417) as a section devoted to potential caveat, and another one envisioning future studies and recommendations capable to amend and bridge these limitations.

Answer: We sincerely thank the reviewer for constructive criticisms and valuable suggestions. We discussed the limitations of transcriptome analysis, as paralogous variation in the discussion section.

Line 496-511

“Although transcriptome sequencing has the convenience of quickly obtaining a large amount of gene expression information, it still has certain limitations, such as paralogs variation. Most plants have undergone multiple rounds of whole genome duplication (WGD) [45], which has long been recognized as an important evolutionary force. It is important to know whether there are correlations between expression divergences of paralogs. Several positive correlations between the divergence of gene and protein expression were identified in mammals. However, a study in sunflower has indicated that there are no correlations [46]. In this study, two unigenes (Cluster-11764.13220 and Clus-ter-11764.7792) were identified as homologs of leucine-rich receptor-like protein kinase family protein AT1G17230. The expression of these two genes is significantly positively correlated, with high expression in February and no expression detected in the samples from May to September (Table 3). However, there are certain differences in gene expression homologs (Cluster-11764.8337 and Cluster-11764.12867) of leucine-rich repeat transmembrane-type receptor kinase GSO1. Cluster-11764.8337 is expressed in February, while Cluster-11764.12867, although expressed at the highest level in February, is expressed between March and September (Table 3). Functional divergences and expression differences of paralogs have arisen due to selective pressures throughout evolution [46,47]. Hence, a comprehensive investigation of the expression patterns of paralogous gene pairs in response to various stresses and a study of correlations between the expression levels and sequence divergences of the paralogs are needed in further analysis.”

  1. I also could not help noticing that authors try performing a metabolic network/pathway synthesis if figure 4. Please complement this with a pathways-enriched analysis as carried out in Agronomy2023 13:1396 (details in the second paragraph of section 2.9, page 7).

Answer: We sincerely thank the reviewer for constructive criticisms and valuable suggestions. We added new Table 2 to show the top KEGG pathways enriched by differentially expressed genes (DEGs) and added KEGG enrichment of DEGs to method and result section.

Line 193-199

“2.6. KEGG Enrichment Result of DEGs

The top five KEGG pathways enriched by DEGs of clusters are represented in Table 2. The KEGG pathway “Metabolic pathways” (ko01100) and “Biosynthesis of secondary metabolites” (ko01110) were enriched by DEGs of C9-C11 and C4-C8, “Carbon metabolism” (ko01200) was annotated by 57 over-expressed genes at March and May (cluster C4-C8), and 54 up-regulated genes of C1 were annotated in the KEGG pathway “Biosynthesis of amino acids” (ko01230). Besides, “Fatty acid degradation” (ko00071) was enriched by genes commonly up-regulated in August (cluster C3-C7).”

Line 656-660

“4.8. GO and KEGG Enrichment

We employed the topGO toolkit within R for GO enrichment analysis of DEGs and various module genes. The DEGs of elm unigene IDs were transferred to the Arabidopsis TAIR locus IDs during the MapMan analysis. The software KOBAS were used to test the statistical enrichment of differential expression genes in KEGG pathways in elm.”

  1. On the other hand, authors refer in several occasions to transcriptional responses related to abiotic stress (in L226 L410 L551). Are there any particular hints on some gene families that have previously been linked with abiotic stresses tolerance? For instance, refer to: (i) Plant Science 2016 242:250 for the ERECTA gene family in association with the AP2 domain, (ii) BMC Genetics 2012 13:58 for the ASR family f in association with the ABA-dependent MYB, and (iii) Theor Appl Genet 2012 125(5):1069-85 for the DREB transcription factor family pleotropic in several pathways with the WRKY transcription factor, all of the above revisited at Genes 2021 12(4):556. I recommend authors to discuss and refer to these cases. Please revisit this point during the discussion section. Other target candidate genes worth exploring and referring to as part of the discussion on abiotic stress tolerance are enlisted in Front Plant Sci 2018 9:128 and Front Genet 2019 10:954.

Answer: We sincerely thank the reviewer for constructive criticisms and valuable suggestions. We added the potential effects of some gene families (ASR, DREB2 and ERECTA) that have previously been linked with abiotic stresses tolerance to the discussion section. Besides, we added new Table 3 to show expression patterns of genes participating in the drought stress responding. The literature mentioned by the reviewer has been cited in the references.

Line 309-324

“The annual rainfall and the number of drought stress gene expression indicate that drought stress continues to affect the growth and development of elm trees between February and June. The genes of cluster C9 and C11 are involved in transcriptional regulation and plant hormone response, which indicated these processes are related to drought stress responding. ERECTA genes function as transcriptional regulators and are among the best characterized genes affecting drought tolerance [34,35]. Six homologs of ERECTA were identified in elm, these genes all had the highest expression levels in February (Table 3). The transcription factors of the Asr (abscisic acid, stress, ripening induced) family of genes are plant-specific and stress-regulated components of the ABA- dependent pathway [36,37]. In elm, the homolog of Asr highly expressed at February and June. Besides, drought-responsive element binding (DREB) protein encoding Dreb genes are plant-specific, stress regulated transcription factors which belong to the AP2/ EREBP fam-ily and which are in the ABA-independent pathway for drought stress response [38]. In this study, three homologs (Cluster-11764.29311, Cluster-11764.8458 and Clus-ter-11764.22853) of DREB2 were found highly expressed at February. These results suggested that these genes in elm possibly played analogous roles in drought stress respond-ing.”

  1. Last but not least, please also embrace in the novel perspectives section at L417 how could the identified expression patterns source the understanding of seasonal branch growth in other tree species besides elm? Specifically, a major question that authors should prospect in their discussion is how to unlock and effectively utilize these novel gene functionalities. Speed breeding, gene editing, recurrent backcrossing and inter-specific schemes (condensed, contrasted and discussed in Trends Genet 2021 37:1124-1136, and Front Plant Sci 2020 11:583323, please refer to) may offer some insights. Another broader reflection to embrace in this novel perspective section is the interplay of seasonal branch growth and adaptation to abiotic pressures and habitat gradients in an eco-physiological context (refer to the seminal reviews Front Genet 2020 11:564515 and Front Genet 2022 13:910386).

Answer: We sincerely thank the reviewer for constructive criticisms and valuable suggestions. We added the prediction of the necessary next step to understand the genetic basis of phenotype in elm, as CRISPR system and population genomics analysis to the discussion section. The literature mentioned by the reviewer has been cited in the references.

Line 521-529

“Advances in genome sequencing technologies combined with efficient trait mapping pro-cedures accelerate the availability of beneficial alleles for breeding and research. Once we have evidence of candidate genes involved in the adaptation or development of elm, the necessary next step to understand the genetic basis of these phenotype, is to design in vitro systems, knock-out, knock-down or knock-in experiments that validate the involvement of such genes in the observed phenotypes [48,49]. Repeats (CRISPR) system alongside the CRISPR associated protein 9 (Cas9), commonly known as CRISPR/Cas9, which can be used to eliminate, introduce or replace specific segments of DNA within a targeted site in a genome.”

Line 530-546

“Elm is a tree species with outstanding drought resistance, and there are many elm populations distributed in the Inner Mongolia Autonomous Region in China, which has low annual rainfall in China. Characterizing the population history of elm and identifying loci underlying local adaptation is crucial in functional ecology, evolutionary biology, conservation and agronomy. Population genomics now provides tools to better integrate selection into a historical framework, and take into account selection when reconstructing demographic history [47,50]. Although there is currently no reported whole genome se-quencing data for elm, we believe that we will obtain its whole genome data in the near future. Transcriptomics provides windows into molecular variation in breeding lines, these windows are closer to phenotype. Complex quantitative trait loci (QTLs) and ge-nome-wide association studies (GWAS) can provide independent sets of markers to com-plement genetic markers as breeding tools. Even if we don't have the genome of the elm tree, SNP detection by RNA-seq is particularly interesting for non-model model organisms. SNPs detected in expressed regions can be used to characterize variants affecting protein functions, and to study cis-regulated genes by analyzing allele-specific expression (ASE) in the interplay of seasonal branch growth and adaptation to abiotic pressures and habitat gradients in elm.”

Reviewer 3 Report

Understanding the molecular mechanisms beyond wood growth is a highly necessary aim in the context of climate changes and carbon sequestration, for which I congratulate the authors. Yet, I cannot recommend a transcriptomic study based on a single individual plant (either here or any other journal). I do understand that the tree was followed during 2 years and that technical samplings were performed on February, March, May, 471 June, August, and September. Yet, this simply misses the variation that occurs between biological replicates. Biological replicates are important because they address how widely your experimental results can be generalized. They indicate if an experimental effect is sustainable under a different set of biological variables. For instance, because the tree comes from a natural environment, the authors cannot rule out other intrinsic processes as pathogens, diseases, stress, which are often not visible. That is why biological replicates are necessary.

See above.

Author Response

Responses to Reviewer 3’s comments:

Understanding the molecular mechanisms beyond wood growth is a highly necessary aim in the context of climate changes and carbon sequestration, for which I congratulate the authors.

We sincerely thank the reviewer for constructive criticisms and valuable suggestions. For improving the quality of the manuscript, we have made the following modifications to the manuscript:

For figures, we modified Figure1: added new panel B-C to show the other two trees used for analysis. We chose three robust 15-year-old elm trees as three reduplicates for our study. We added Figure S3 to show the the consistency of gene expression in samples collected from three trees, 50 genes were selected for RT-PCR experiments in this analysis.

For tables, we added new Table 2 to show the top KEGG pathways enriched by differentially expressed genes (DEGs). Besides, we added new Table 3 to show expression patterns of genes participating in the drought stress responding.

For manuscript, we rewrote the abstract to focus on key findings and report the sampling genotyping details. We added KEGG enrichment result of DEGs to result section. We added three paragraphs to the discussion section: (1) potential effects of some gene families (ASR, DREB2 and ERECTA) that have previously been linked with abiotic stresses tolerance; (2) limitations of transcriptome analysis, as paralogous variation; (3) prediction of the necessary next step to understand the genetic basis of phenotype in elm, as CRISPR system and population genomics analysis. The sampling genotyping details of three elm trees, and KEGG enrichment analysis were added to the method section.

Comments:

1 Yet, I cannot recommend a transcriptomic study based on a single individual plant (either here or any other journal). I do understand that the tree was followed during 2 years and that technical samplings were performed on February, March, May, 471 June, August, and September. Yet, this simply misses the variation that occurs between biological replicates. Biological replicates are important because they address how widely your experimental results can be generalized. They indicate if an experimental effect is sustainable under a different set of biological variables. For instance, because the tree comes from a natural environment, the authors cannot rule out other intrinsic processes as pathogens, diseases, stress, which are often not visible. That is why biological replicates are necessary.

Answer: We sincerely thank the reviewer for constructive criticisms and valuable suggestions. We sincerely apologize for the mistake we made in writing the sampling method. We chose three robust 15-year-old elm trees as three biological reduplicates for our study. We modified the sampling in the method section, added new panel B-C of Figure 1 to show the other two trees (tree2 and tree3) used for analysis. Besides, to show the consistency of gene expression in samples collected from three biological reduplicates, 50 genes were selected for RT-PCR experiments in this analysis. In most cases, the gene expression trends were similar among three elm trees, the correlation between the tree1 and tree2 was cor = 0.692, the correlation between the tree1 and tree3 was cor = 0.911. The correlation test results and expression patterns of 7 representative genes in the three tree samples are shown in new Figure S3.

Line 550-563

“We chose three robust 15-year-old elm trees for our study, the cultivar used in this study were collected from the Inner Mongolia Autonomous Region in China (39°12′32″ N and 101°38′36″ E). The collected seeds of the cultivar were planted and nurtured in the tree farm of Shandong Normal University in Shandong Province, China, with geographical coordinates 36°32′45″ N and 116°50′2″ E (Figure 1A-C). Monthly samples were taken from the tree from February to September, the sampling date is generally the 15th of each month. For the purpose of our research, we focused on 2-year-old elm branches, which were subject to morphological, physiological, and transcriptomic examinations.”

Line 597-598

“In transcriptome analysis, samples from three trees were treated as three biological replicates.”

Line 693-697

“To verify the consistency of gene expression in samples collected from three trees, we se-lected 50 genes for RT-PCR experiments. These genes are involved in processes such as “translation elongation”, “mitosis”, “transcriptional regulation”, “cell wall synthesis”, “sucrose synthesis”, and “lipid synthesis”, as well as 13 randomly selected genes.”

Line 229-238

“To verify the expression similarity of genes among three elm trees, 50 unigenes were selected for validation. Primers were designed to span exon–exon junctions (Table S7). In most cases, the gene expression trends were similar among three elm trees, the correlation between the tree1 and tree2 was cor = 0.692, the correlation between the tree1 and tree3 was cor = 0.911. The correlation test results and expression patterns of 7 representative genes in the three tree samples are shown in Figure S3. For example, the homolog of Secondary cell wall biosynthesis participating gene CESA4, Cluster-11764.29560, similar expression patterns of this gene were observed in the branch samples from three elm trees: upregulat-ed expression began in March, reached its highest level in May, and then began to decline (Figure S3).”

Reviewer 4 Report

The study is interesting, merit, and well-prepared. However, why the authors have selected only one single Ulmus species and did not analyze other tree species/genera?

According to my knowledge, Ulmus is not a dominant tree species in forest ecosystems, so the possibility of applying the results of these studies in forestry is low. This study would be far more interesting if prepared on several different tree species (i.e. several different tree genera), preferably those, which grow together in the forest ecosystems and are economically important to the regional/continental forestry. 

Lines 221-222
Authors' wrote:
"Elm, with its timber being commercially significant, especially in tropical nations, offers multiple applications across industries."

Please, refer to the "commercially significant" value of Siberian ulm timber.

Admittedly, the wood of some elm species (e.g. Ulmus parvifoliaUlmus minor) is used due to its hardness and strength. However, according to my knowledge, Ulmus pumila is used as an ornamental tree (or rather, ornamental shrub), not in wood production. 

Moreover, were similar studies conducted on Fagales trees, such as oak and beech species? Fagales trees are widely used in wood production in the Nothern Hemisphere and belong to the major deciduous tree species in forest ecosystems in North America, Europe, and Asia.

I strongly recommend you either, use the presented method on Fagales species or involve the results of studies on Fagales trees, especially oaks, if available. It increases the application value of the paper in forestry. 

Lines 220-221
Authors wrote: "Lately, comprehensive transcriptomic analyses of wood formation in Populus and Pinus have been brought to light."

If I understand correctly, there are no other studies on the transcriptomic analyses of wood, but those conducted on pine and poplar wood. Is that true? If yes, please expand it and explain the reason, why the presented method has not found application in forestry/wood science.

In addition, please, add "Siberian" to the title of the manuscript. The English name of Ulmus pumila is "Siberian elm".

none

Author Response

Responses to Reviewer 4’s comments:

The study is interesting, merit, and well-prepared. However, why the authors have selected only one single Ulmus species and did not analyze other tree species/genera?

According to my knowledge, Ulmus is not a dominant tree species in forest ecosystems, so the possibility of applying the results of these studies in forestry is low. This study would be far more interesting if prepared on several different tree species (i.e. several different tree genera), preferably those, which grow together in the forest ecosystems and are economically important to the regional/continental forestry.  

We sincerely thank the reviewer for constructive criticisms and valuable suggestions. For improving the quality of the manuscript, we have made the following modifications to the manuscript:

For figures, we modified Figure1: added new panel B-C to show the other two trees used for analysis. We chose three robust 15-year-old elm trees as three reduplicates for our study. We added Figure 6 to show the predicted regulatory relationship of homologs of xylem development gene in P. trichocarpa by extracting co-expression relationships from WGCNA network. We added Figure S3 to show the the consistency of gene expression in samples collected from three trees, 50 genes were selected for RT-PCR experiments in this analysis.

For tables, we added new Table 2 to show the top KEGG pathways enriched by differentially expressed genes (DEGs). Besides, we added new Table 3 to show expression patterns of genes participating in the drought stress responding.

For manuscript, we rewrote the abstract to focus on key findings and report the sampling genotyping details. We added KEGG enrichment result of DEGs to result section. We added three paragraphs to the discussion section: (1) potential effects of some gene families (ASR, DREB2 and ERECTA) that have previously been linked with abiotic stresses tolerance; (2) the correlation analysis between the expression data of elm and single-cell transcriptome of in P. trichocarpa; (3) limitations of transcriptome analysis, as paralogous variation; (4) prediction of the necessary next step to understand the genetic basis of phenotype in elm, as CRISPR system and population genomics analysis. The sampling genotyping details of three elm trees, and KEGG enrichment analysis were added to the method section.

Comments:

1 Lines 221-222

Authors' wrote: "Elm, with its timber being commercially significant, especially in tropical nations, offers multiple applications across industries."

Please, refer to the "commercially significant" value of Siberian ulm timber..

Admittedly, the wood of some elm species (e.g. Ulmus parvifoliaUlmus minor) is used due to its hardness and strength. However, according to my knowledge, Ulmus pumila is used as an ornamental tree (or rather, ornamental shrub), not in wood production. 

Answer: We sincerely thank the reviewer for constructive criticisms and valuable suggestions. We modified this sentence. Please find:

Line 250-251

“Siberian elm, is used as an ornamental tree/shrub, especially in temperate countries, offers multiple applications across landscaping.”

2 Moreover, were similar studies conducted on Fagales trees, such as oak and beech species? Fagales trees are widely used in wood production in the Nothern Hemisphere and belong to the major deciduous tree species in forest ecosystems in North America, Europe, and Asia.

I strongly recommend you either, use the presented method on Fagales species or involve the results of studies on Fagales trees, especially oaks, if available. It increases the application value of the paper in forestry.

Answer: We sincerely thank the reviewer for constructive criticisms and valuable suggestions. For increases the application value of this study, we conducted correlation analysis between the expression data of elm and single-cell transcriptome of in P. trichocarpa. We added Figure 6 to show the predicted regulatory relationship of homologs of xylem development gene in P. trichocarpa by extracting co-expression relationships from WGCNA network. Please find:

Line 675-684

“5.0. Correlation Analysis of Xylem Cell Development Between Elm and P. trichocarpa

By integrating scRNA-seq-lcmRNA-seq correlation, scUPlcmUP gene distribution and known gene functions, Tung Chia-Chun et al. revealed the single-cell clusters of vessel el-ements (Ptr1), libriform fbers (Ptr7), and ray parenchyma cells (Ptr8) in P. trichocarpa [45]. Based on the homologous relationship between poplar and elm genes, we firstly identified the homologous genes of the genes in the three clusters (Ptr7, Ptr1, Ptr8) in elm, and further screened out the genes potentially involved in the above three processes in elm through the expression patterns in May and June. By assessing the interlinked co-expression rela-tions between genes, we were also able to project several potential genes that might be in-tertwined with established genes vital for xylem cell development.”

Line 471-490

“The development of individual xylem cell types has mostly been studied through an-atomical observations over the past hundred years, Tung Chia-Chun et al unraveled the developmental processes of plant xylem through single-cell RNA sequencing (scRNA-seq) analyses. We conducted correlation analysis between the expression data of elm and sin-gle-cell transcriptome of in P. trichocarpa, the result was shown on Figure 6. In poplar, genes of libriform fibers (Ptr7) are heavily enriched with SCW biosynthesis genes, especial-ly those for monolignol biosynthesis [45]. Ray parenchyma cells (Ptr8) have moderate ex-pression of monolignol biosynthesis genes, consistent with previous observations that ray parenchyma cells in Picea also contribute to lignification [46]. In elm, the homologs of lig-nin formation and the primary phenylpropanoid route up-regulated and co-expressed in May, as PtrHCT1 (HCT in A. thaliana), PtrPAL3 (HCT), PtCesA4 (CESA4), PtrCAld5H1(FAH1) and PtrCslA2 (ATCSLA09) (Figure 6A). These results provided important insights into sim-ilarities in developmental trajectories between poplar and Siberian elm. In plants, cell po-larization is the specialization of developmental events along one orientation or one direc-tion. In model plants, receptor-like transmembrane kinase I (TMK1) is the key regulator in auxin signaling, TMK1-mediated auxin signalling regulates cell polarity formation and growth of root the apical hook [47,48]. In poplar, homologous of TMK1 are related to vessel elements (Ptr1); in elm, homologous of TMK1 co-expressed with multiple cell wall devel-opment genes (HCT, C3H, TBL19), suggesting the potential regulatory roles of TMK1 in cell wall vessel elements.”

3 Lines 220-221

Authors wrote: "Lately, comprehensive transcriptomic analyses of wood formation in Populus and Pinus have been brought to light."

If I understand correctly, there are no other studies on the transcriptomic analyses of wood, but those conducted on pine and poplar wood. Is that true? If yes, please expand it and explain the reason, why the presented method has not found application in forestry/wood science.

Answer: We sincerely thank the reviewer for constructive criticisms and valuable suggestions. We modified this sentence. Please find:

Line 246-250

“Yet, our comprehension of the foundational molecular activities related to vascular cambium onset, vascular structuring, and xylem differentiation remains rudimentary. Lately, comprehensive transcriptomic analyses of wood formation in Populus, Pinus, white teak (Gmelina arborea Roxb), Eucalyptus grandis, and Liriodendron chinense. have been brought to light. ”

4 In addition, please, add "Siberian" to the title of the manuscript. The English name of Ulmus pumila is "Siberian elm".

Answer: We sincerely thank the reviewer for constructive criticisms and valuable suggestions. We modified the title. Please find:

“Comprehensive time-course transcriptome reveals the crucial biological pathways involved in the seasonal branch growth in Siberian Elm (Ulmus pumila)”

Round 2

Reviewer 2 Report

This revised version had been carefully edited and the overall work has substantially improved. I must confess that readability is much better, now with a clearer hypothesis-oriented framework. I can recommend acceptance at the present stage.